# Fast retrievals of tropospheric carbonyl sulphide with IASI

R. Anthony Vincent[1] and Anu Dudhia[1]

[1]Atmospheric, Oceanic, and Planetary Physics, Oxford University, Clarendon Laboratory, Parks Road, Oxford OX1 3PU, UK

*Correspondence to:* R. Anthony Vincent (tony.vincent@physics.ox.ac.uk)

**Abstract.** Iterative retrievals of trace gases, such as carbonyl sulphide (OCS), from satellite can be exceedingly slow. The algorithm may even fail to keep pace with data acquisition such that analysis is limited to local events of special interest and short time spans. With this in mind, a linear retrieval scheme was developed to estimate total column amounts of OCS at a rate roughly $10^4$ times faster than a typical iterative retrieval. This scheme incorporates two concepts not utilised in previously published linear estimates. First, all physical parameters affecting the signal are included in the state vector and accounted for jointly, rather than treated as effective noise. Second, the initialisation point is determined from an ensemble of atmospheres based on comparing the model spectra to the observations, thus improving the linearity of the problem. The entirety of 2014 data from the Infrared Atmospheric Sounding Interferometer (IASI), instruments A and B, were analysed and showed spatial features of OCS total columns including depletions over tropical rainforests, seasonal enhancements over the oceans, and distinct OCS features over land. Error due to assuming linearity was found to be on the order of 11% globally for OCS. Comparisons to surface volume mixing ratio *in situ* samples taken by NOAA show seasonal correlations greater than 0.7 for five out of seven sites across the globe. Furthermore, this linear scheme was applied to OCS, but may also be used as a rapid estimator of any detectable trace gas using IASI or similar nadir-viewing instruments.

## 1 Introduction

Retrieving atmospheric trace gas concentrations from infrared satellite observations can be an expensive process, especially when implementing an inverse method such as optimal estimation (Rodgers, 2000). In this approach, a radiative transfer model (RTM) describing the physics of light propagating through the atmosphere is iteratively evaluated for every pixel while comparing the model spectrum of the estimate to the measurement. Constraints upon the solution are generally required when estimating more parameters than are independently represented in the observation. While such methods approach theoretical limits of detectability, iteratively evaluating the RTM can be such a time consuming process that the retrieval fails to keep pace with data acquisition. In the case of nadir viewing satellite instruments collecting over $10^6$ observations per day, like the Infrared Atmospheric Sounding Interferometer (IASI) described in Hilton et al. (2011), the computational deficit can limit retrievals to event or region specific analysis. Therefore, fast retrieval methods are required (in the absence of scalable computer clusters) for larger data projects, such as global seasonal analysis, until more significant advances in RTM speed and computational power are realized.

This paper presents a new method for rapidly retrieving trace gas abundances as applied towards estimating total vertical column amounts of carbonyl sulphide (OCS). The proposed method is linear in the sense that an estimate for each pixel is made only once, thus bypassing the iterative steps. By pre-calculating the RTM the retrieval operates roughly $10^4$ times faster than an iterative method using a line-by-line RTM, even when performance is sped up using monochromatic look-up tables (Vincent and Dudhia, 2016). Improving retrieval speed this dramatically comes at the cost of reduced accuracy compared to more robust iterative methods. Therefore, retrievals focused on individual pixels around specific scenes are best left for optimal estimation techniques, like the retrieval developed by Liuzzi et al. (2016), while the method presented here is useful for rapid monthly to seasonal analysis where modest averaging reduces random error to improve the quality of the mean or median value.

The approach presented here differs from previous work on fast linear retrievals in two ways: First, an initialisation point is selected from an ensemble of atmospheres based upon how closely the corresponding model spectrum matches the observed spectrum. Previous work generally uses a global or a wide region mean atmosphere as the initial guess. By selecting from an atmospheric ensemble, the problem becomes more linear and reduces the non-linear error introduced by failing to iterate towards a converged solution. Second, all physical parameters affecting the spectral signal above instrument noise are jointly estimated to account for their influence upon the desired quantity (OCS total columns in this case). One popular alternative, as first described by von Clarmann et al. (2001), is to create an effective measurement covariance that treats the components of the signal due to variations in parameters not explicitly retrieved as noise (Walker et al., 2011; Van Damme et al., 2014). The two methods (joint retrievals versus effective noise) produce identical results under idealised conditions. However, in practice jointly retrieving noticeable physical parameters is expected to perform better, but with a greater number of algebraic operations per estimate.

Atmospheric OCS estimates from IASI observations throughout 2014 are used as a case study for this new rapid retrieval method, because OCS is an important trace gas towards understanding the global sulphur cycle, is currently poorly modelled, and is at the edge of detectability with nadir viewing instruments like IASI. While OCS is studied here, the proposed retrieval method can potentially be used for any detectable trace gas. Aside from introducing a novel retrieval method, this paper also shows unprecedented seasonal OCS results from a nadir-viewing hyperspectral instrument.

## 2 Carbonyl sulphide (OCS): Background

Carbonyl sulphide is a molecular reservoir species for atmospheric sulphur. OCS is the longest lived and most abundant sulphur containing gas in the unpolluted atmosphere (Notholt et al., 2006). Therefore, knowledge of OCS distributions, sources, and sinks are crucial towards understanding the global sulphur cycle. Furthermore, OCS photochemically converts to sulphate aerosols once vertically convected towards the stratosphere, thus affecting global climate by scattering incoming short-wave radiation. In fact, previous work suggests that OCS is the primary source of stratospheric sulphates during periods of low volcanic activity (Notholt et al., 2003).

## 2.1 Sources and sinks

Yearly OCS trends are approximately constant according to numerous NOAA sample stations across the globe (Montzka et al., 2007). However, Kremser et al. (2015) detected a slight increase in both tropospheric and stratospheric OCS since 2001 of 0.5–1.0% per year over sites in Australia and New Zealand. Historically, OCS is approximately 25–40% greater in concentration today than it was during pre-industrial times (Aydin et al., 2002; Montzka et al., 2004). Nonetheless, current data show that global sources of OCS generally balance the sinks in the near term (past three decades).

The majority of OCS originates from ocean sources either by direct emissions or secondary production from short-lived oceanic $CS_2$ or dimethyl sulphide (DMS) gas (Barnes et al., 1994). The proportions and mechanisms of these sources are still largely uncertain. However, Launois et al. (2015a, b) proposed a new model suggesting that direct OCS ocean emission from photochemically reduced chromophoric dissolved organic matter (CDOM) is dominant. The amount of OCS released from this process is then a function of CDOM concentrations near the surface, water clarity, and incident ultra-violet radiation. While $CS_2$ may arise from numerous sources, including photochemical reduction of CDOM, DMS is overwhelmingly a product of living oceanic phytoplankton (Sunda W. et al., 2002).

The remaining sources of OCS are largely anthropogenic with a small contribution from anoxic soils, such as marshes and wetlands. Industrial production of rayon and cellophane are known to emit $CS_2$, where the majority converts to OCS on the order of days. Combustion of sulphur-heavy fossil fuels from coal power plants, petrol (gasoline) engines, and diesel engines also produce OCS and $CS_2$ as by-products. Another substantial anthropogenic source of OCS are oil refineries and natural gas facilities that attempt to remove dissolved sulphur compounds (mostly $H_2S$) for air quality management. This greatly reduces $SO_2$ production during combustion, but OCS and $CS_2$ are created during the recovery operation which may leak into the atmosphere (Chin and Davis, 1993).

The vast majority (over 80%) of OCS is removed from the atmosphere in conjunction with photosynthesis, either from vegetative canopy or microscopic organisms in oxic soils, e.g., *Mycobacterium*. OCS takes the same diffusive pathway as $CO_2$ through plant stomata to the reaction sites in the chloroplasts, where it then reacts with the enzyme carbonic anhydrase (CA) and $H_2O$ to split OCS into $CO_2$ and $H_2S$ (Protoschill-Krebs and Kesselmeier, 1992). Ingestion of OCS via leaf uptake is a one way ticket, meaning, plants do not respire OCS as they do unused $CO_2$. Since OCS is roughly four times more variable than $CO_2$, Berry et al. (2013) suggested that remote detection of OCS could be used as a proxy towards estimating $CO_2$ fluxes over areas of dense vegetation. The remaining portion of the OCS sink budget is atmospheric loss due to reaction with the $OH^-$ and $O^-$ radicals along with stratospheric photolysis (Kettle et al., 2002).

## 2.2 Previous estimates from satellite

Aside from the Atmospheric Trace MOlecule Spectroscopy (ATMOS) experiment using manned space flight, OCS was first observed from satellite by the Interferometric Monitor for Greenhouse Gases (IMG) (Clarisse et al., 2011). Since IMG collected data for less than two years (1996-1997), satellite remote sensing of OCS was not pursued further until the launch of the Atmospheric Chemistry Experiment (ACE) instrument that began operation in 2003 (Barkley et al., 2008). ACE is a solar

occultation instrument that views the sun through the limb of Earth's atmosphere. Therefore, ACE is well designed for strato-spheric sensitivity, but cannot reliably sound the troposphere below an altitude of approximately $8.5\,\mathrm{km}$. There were two major results from this work. First, they showed that OCS vertical profiles above the tropopause decrease steadily with altitude, thus confirming that there is no appreciable source of OCS due to stratospheric chemistry. Secondly, stratospheric OCS tends to be greater towards the equator and less at the poles. This general trend was also confirmed for tropospheric OCS based on a compilation of zenith-viewing ground observations and balloon campaigns (Krysztofiak et al., 2015).

In retrospective analysis, Glatthor et al. (2015) used the limb-viewing Michelson Interferometer for Passive Atmospheric Sounding (MIPAS) instrument to retrieve OCS concentrations at the lowest-most detectable level, $250\,\mathrm{hPa}$, using standard optimal estimation techniques. The compiled results from $2002 - 2012$ in $5°$ by $15°$ latitude-longitude bins showed clear evidence of elevated ocean sources and tropical rainforest sinks that vary with season. However, limb-viewing instruments are not ideal for tropospheric sounding and the $250\,\mathrm{hPa}$ level fails to probe the troposphere at high latitudes as the tropopause decreases in altitude from the equator.

Most recently, Kuai et al. (2014) developed an optimal estimation retrieval scheme to estimate OCS amounts using the Tropospheric Emissions Sounder (TES). TES is a nadir-viewing Fourier transform spectrometer (FTS) instrument aboard NASA's Aura satellite that was launched into polar orbit in 2004. TES is similar in many ways to IASI, but with finer spectral and spatial resolution. However, since TES does not cross scan transverse to its orbital path like IASI, the spatial coverage of TES is much less in comparison.

This retrieval first estimates a vertical profile of OCS on many vertical levels and then averages the levels between $900$ and $200\,\mathrm{hPa}$, because the degrees of freedom for the signal (DFS) of the profile is less than one when using a prior constraint of $20\%$ OCS variability. The DFS is qualitatively defined to be the number of independent pieces of information that come from the signal rather than the noise (Rodgers, 2000, ch. 2.4). Therefore, only one bulk level of OCS is ever distinguishable and even then it is a weighted combination of the true estimate and the *a priori*, which was taken by Kuai et al. to be spatially flat across all locations. The OCS retrieval is carried out after the routine retrieval of temperature, $H_2O$, $O_3$, CO, $CO_2$, $CH_4$, surface temperature, emissivity, cloud optical depth, and cloud pressure. Only scenes with a cloud optical depth less than $0.5$ are considered as cloudy scenes further reduce the OCS information content. The OCS retrieval itself then jointly includes $CO_2$, $H_2O$, surface temperature, cloud optical depth, and cloud pressure in the state vector and uses the posterior covariances from the preprocessed retrieval as the constraints for these extra parameters in the OCS retrieval.

A monthly mean of TES OCS results from June 2006 was published in Kuai et al. (2015), which further validated the concept that direct ocean emissions of OCS are much greater than previously thought (Berry et al., 2013). The published data included retrievals over ocean between $\pm40°$, because the DFS rapidly fell to values less than $0.5$ outside of this range. This means that the majority of the estimates at higher latitudes were dominated by the flat prior OCS field rather than the true OCS concentrations. To put it another way, the uncertainties from an unconstrained retrieval outside of this latitude range would be greater than the prior constraint of $20\%$ variability. An alternative approach would be to lessen the prior OCS constraint to extend the detectable latitude range, but at the expense of greater uncertainty in the retrieved values. However, the increase in uncertainty can be mitigated by averaging over a greater number of pixels which reduces uncertainty by the square root of the

sample size. On the other hand, if the retrieved estimates are mostly *a priori* from tight systemic constraints, then no amount of averaging changes this fact. Nonetheless, the TES product created by Kuai et al. is the current leading retrieval scheme of tropospheric OCS.

## 3 Method description

5 This section methodically discusses the mathematical framework, formulation, and parameter validation of the retrieval scheme applied to OCS. Caution is advised to not overly compare the presented method to a standard optimal estimation routine based upon iterating a time consuming forward model. The intent of this method is to rapidly estimate OCS in a single step with minimal dependence upon prior assumptions. Retrieval error due to avoiding the residual non-linearities are statistically quantified for reference.

### 3.1 Linear retrieval framework

A forward model ($F$) is a numerical construct that represents the physics of how a given state produces an observable quantity. In this case, $F$ models how electromagnetic radiation propagates through an atmospheric state ($\boldsymbol{x}$) to yield the radiance observed by IASI ($\boldsymbol{y}$) with $m$ number of spectral channels. The work presented in this paper uses the Reference Forward Model (RFM) to simulate such spectral radiances (Dudhia, 2016). When the radiative transfer function is sufficiently linear about a
15 reference state vector ($\boldsymbol{x_0}$) of length $n$, $F$ can be linearised according to

$$\boldsymbol{y} - F(\boldsymbol{x_0}) = \mathbf{K}(\boldsymbol{x} - \boldsymbol{x_0}) + \boldsymbol{\epsilon}. \tag{1}$$

Here $\boldsymbol{\epsilon}$ is the error in the measured signal relative to the linearised forward model and $\mathbf{K} \in \mathbb{R}^{m \times n}$, referred to as both the "weighting function" and the "Jacobian", is defined to be a matrix of partial derivatives such that $K_{ij} = \partial F_i(\boldsymbol{x}) / \partial x_j$.

Solutions to Eq. (1) can be estimated in the optimal estimation framework by considering a linearisation about an *a priori*
reference state ($\boldsymbol{x}_\mathrm{a}$). Estimates of an atmospheric state ($\hat{\boldsymbol{x}}$) are given by

$$\begin{aligned}
\hat{\boldsymbol{x}} &= \boldsymbol{x}_\mathrm{a} + \left(\mathbf{K}^\mathrm{T}\mathbf{S}_\epsilon^{-1}\mathbf{K} + \mathbf{S}_\mathrm{a}^{-1}\right)^{-1}\mathbf{K}^\mathrm{T}\mathbf{S}_\epsilon^{-1}\left(\boldsymbol{y} - F(\boldsymbol{x}_\mathrm{a})\right) \\
&= \boldsymbol{x}_\mathrm{a} + \mathbf{G}\left(\boldsymbol{y} - F(\boldsymbol{x}_\mathrm{a})\right),
\end{aligned} \tag{2}$$

where $\mathbf{G}$ is referred to as the gain matrix (Rodgers, 2000, ch. 4). The covariance matrix of the stochastic error in the measurements is denoted as $\mathbf{S}_\epsilon$. Since raw spectra from a FTS such as IASI are generally uncorrelated, $\mathbf{S}_\epsilon$ has zeroes in the off-diagonal elements while the diagonal elements are the variances of the signal at that spectral position. However, because IASI spectra
are apodized on-board the satellite (Amato et al., 1998), off-diagonal spectral correlations are thus introduced into $\mathbf{S}_\epsilon$. The term *a priori* is meant to include both a mean state, $\boldsymbol{x}_\mathrm{a}$, and its covariance $\mathbf{S}_\mathrm{a}$. Inverting $\mathbf{S}_\mathrm{a}$ in Eq. (2) applies a "soft" constraint upon the solution, penalizing estimates that deviate greatly from the atmosphere provided in the prior estimate.

When the probability density function of the atmospheric state is symmetric about the expected value, the posterior covariance (i.e., the estimated covariance of $\hat{\boldsymbol{x}}$) is found to be

$$\hat{\mathbf{S}}_x = \left(\mathbf{K}^\mathrm{T}\mathbf{S}_\epsilon^{-1}\mathbf{K} + \mathbf{S}_\mathrm{a}^{-1}\right)^{-1}. \tag{3}$$

This is a convenient result, because it means that the uncertainties and correlations between retrieved parameters are generated as a by-product of the retrieval process. Eq. (3) also highlights the fact that if $\hat{\mathbf{S}}_x = \mathbf{S}_a$, then the retrieval has done nothing to improve upon the *a priori* and is completely insensitive to the estimated parameters.

Further diagnostic information about the retrieval is succinctly contained in a unitless $n \times n$ matrix known as the averaging kernel matrix (AKM), defined as

$$\mathbf{A} = \frac{\partial \hat{\boldsymbol{x}}}{\partial \boldsymbol{x}} = \mathbf{GK}. \tag{4}$$

Using this relation, Eq. (2) can be rewritten in the more insightful but less practical form,

$$\hat{\boldsymbol{x}} = (\mathbf{I}_n - \mathbf{A})\, \boldsymbol{x}_a + \mathbf{A}\boldsymbol{x} + \mathbf{G}\boldsymbol{\epsilon}, \tag{5}$$

where $\mathbf{I}_n$ is the identity matrix with $n$ diagonal elements. Written this way, it becomes clear that the estimate of state, $\hat{\boldsymbol{x}}$, is a weighted average of the true state and the prior state. When $\mathbf{A}$ is diagonal, these elements express the fractional proportion of how sensitive the estimate is to the true state. Non-zero values in the off-diagonal elements of $\mathbf{A}$ track the correlation between the estimated parameters within $\hat{\boldsymbol{x}}$. Ideally, $\mathbf{A}$ approaches the identity matrix and no prior state appears in the estimate. However, this is seldom the case for nadir-viewing unless performing a maximum likelihood retrieval where there is by definition no *a priori* information.

Repeated analysis of $\mathbf{A}$ can be unwieldy when developing a retrieval algorithm. Therefore, a scalar "figure of merit" is often desirable that allows for multiple matrices of $\mathbf{A}$ to be compared in a straightforward manner. The DFS, as mentioned in Sect. 2.2, is one such possible metric and is calculated by taking the trace of the averaging kernel matrix,

$$d_s = \text{Tr}\,(\mathbf{A}). \tag{6}$$

Perfectly conditioned non-trivial inverse problems will have DFS values equal to the number of state parameters, $n$.

With these relations at hand, the proposed retrieval is a direct application of Eq. (2), where the RFM is represented by $F$ and used to model IASI radiances and create Jacobian spectra ($\mathbf{K}$). Rather than use a climatological static mean value for $\boldsymbol{x}_a$ as the linearisation point, the ensemble of 80 atmospheres selected to parametrise the RTTOV forward model (Matricardi, 2008) was used to create a subsequent ensemble of initial states ($\boldsymbol{x}_a$), model spectra ($F(\boldsymbol{x}_a)$), and gain matrices ($\mathbf{G}$). The model spectrum that most closely matches the observed IASI spectrum is used to select the initialisation point from the ensemble, which will be discussed further in Sect. 3.6. Once again, the point of this process of selecting a model atmosphere from an ensemble is to make the retrieval as linear as possible without iterating the forward model.

Brightness temperature spectra were intentionally used instead of radiance spectra because removing curvature from the Planck function improves the linearity of the retrieval (Rodgers, 2000, Ch. 5.1). The downside to this is that the measurement noise in brightness temperature space (NE$\Delta$T) becomes a function of the observation (see the red lines in Fig. 5 for example). Therefore, the measurement covariances ($\mathbf{S}_\epsilon$) were adjusted specifically for each atmosphere based on the model spectra when computing the gain matrices. Apodization was modelled in the off-diagonal elements of $\mathbf{S}_\epsilon$ according to the discussion in Amato et al. (1998).

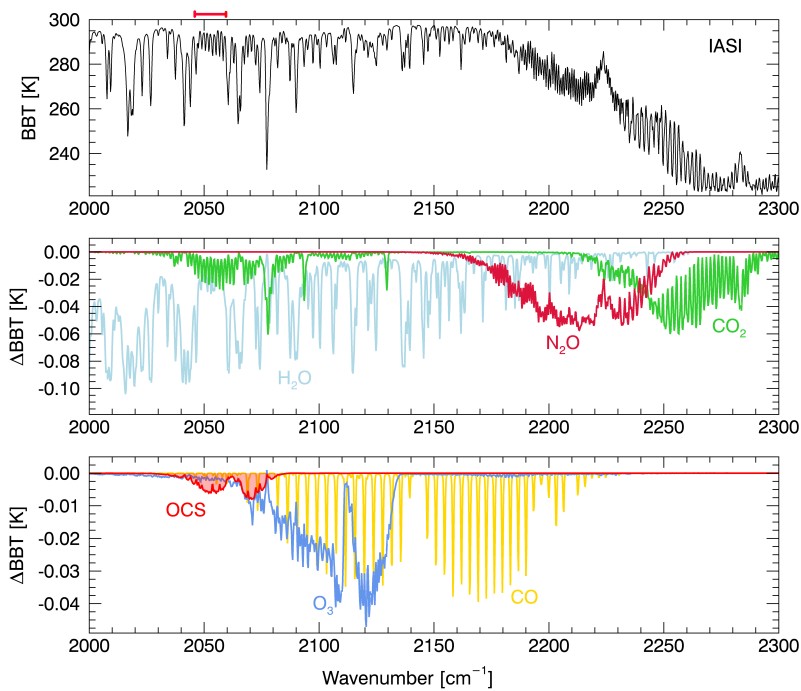

**Figure 1.** Top: A simulated IASI BBT spectrum from a desert (i.e., low humidity) atmosphere covering the spectral range used in the linear retrieval. Middle and Bottom: Jacobian spectra showing the change in BBT for a 1% increase in volume mixing ratio (VMR) for the gases listed. The $CO_2$ and $N_2O$ spectra represent tropospheric perturbations while the remaining four are total column perturbations. The red bar denotes the area between two $H_2O$ lines where a large portion of OCS information comes from.

### 3.2 Spectral range considered

Identifying OCS spectral features is a straightforward process. Figure 1 shows a sample blackbody brightness temperature (BBT) spectrum for the spectral range targeted in this study ($2000 - 2300 \, \mathrm{cm}^{-1}$), including the dominant $\nu_3$ rotational-vibrational band of OCS in the thermal infrared centred at $2060 \, \mathrm{cm}^{-1}$. Notice that $H_2O$ and $CO_2$ are the primary contaminants here with additional contributions from CO and $O_3$. This also shows there are no isolated OCS spectral lines and that the other detectable species must be accounted for explicitly during the retrieval.

The spectral range included in this retrieval is much larger than the OCS spectral band, which runs from $2040 - 2080 \, \mathrm{cm}^{-1}$. This is to provide temperature and contaminating gas information from the spectrum as location specific *a priori* are not used. In particular, the $CO_2$ and $N_2O$ spectral features are of various line strengths which saturate at different effective altitudes throughout the vertical profile. Since these two gases are well mixed with low natural variability, they provide robust information on atmospheric temperature. In an iterative retrieval, a much narrower spectral region would be used and the additional

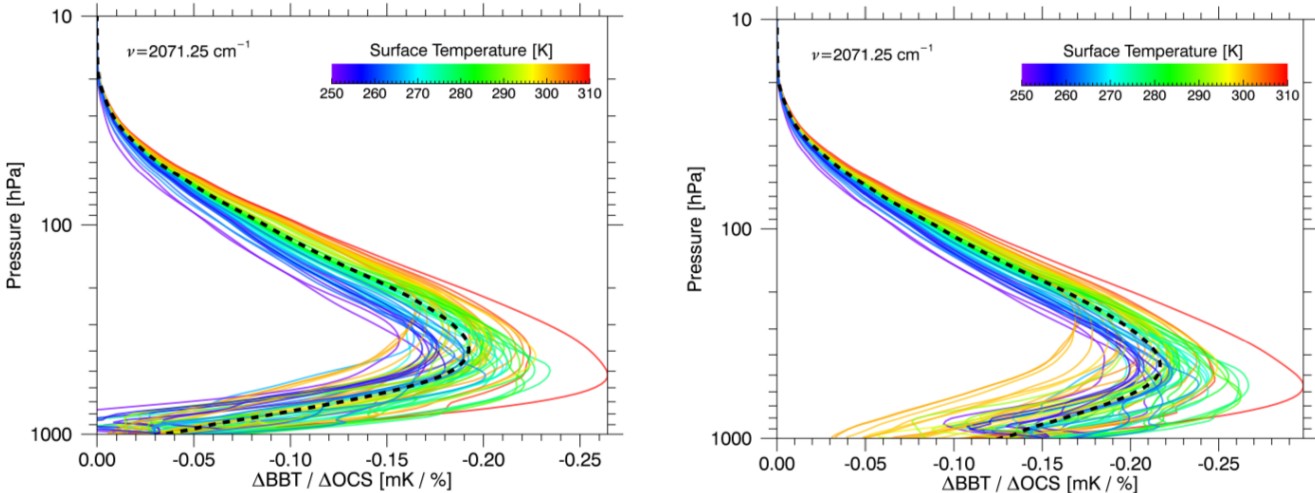

**Figure 2.** Weighting functions for IASI simulations of OCS are shown from vertical layers $1\,km$ thick at $2071.25\,cm^{-1}$ for the 80 atmosphere RTTOV ensemble, which are colour coded by the individual surface air temperature. The left plot represents a ground to surface air thermal contrast scenario of $3\,K$ while the right figure shows a $15\,K$ thermal contrast. The dashed line represents the mean weighting function of the ensemble.

information would be supplanted by weather specific *a priori* to save time computing the forward model. Since the forward model is pre-calculated in this method, the added spectral range only increases the number of linear algebra operations.

The spectral characteristics of the observation and the applied constraints determine the vertical sensitivity of the retrieval. The weighting functions, i.e., the Jacobian values from perturbing each individual vertical layer, for OCS at the strongest

spectral point ($2071.25\,cm^{-1}$) are shown in Fig. 2. Notice that peak OCS sensitivity with IASI is consistently around $500\,hPa$ for all atmospheres and both surface temperature contrast scenarios. This is consistent with the OCS analyses published in Shephard et al. (2009) and Kuai et al. (2014). However, when the surface ground temperature is significantly warmer than the surface air temperature (positive thermal contrast), then lower-most tropospheric OCS becomes up to three or four times more detectable. This is because thermal contrast between the surface and the atmospheric temperature accentuates spectral

absorption or emission features, which makes them easier to distinguish.

### 3.3   Defining the state vector and prior covariance

Even though OCS is the desired target, the intent of the joint retrieval is to simultaneously account for all physical parameters that affect the observed spectrum above the noise level. Mathematically this is handled with the cross-terms in Eq. (2) via the extra rows in **K** when calculating the gain matrix (**G**). If they are not accounted for, then the other physical parameters become

biased into the target estimate. If this were to happen then one could not say with confidence whether an OCS enhancement or depletion was actually due to OCS or something else, such as water vapour or surface temperature. Once the gain matrix

is calculated, only the row of $\mathbf{G}$ corresponding to the desired target (e.g., OCS) needs to be carried through the multiplication of $\Delta \boldsymbol{y}$. However, there is potential diagnostic information resulting from the other jointly estimated parameters to be used for assessing retrieval quality, while computational savings from neglecting all but one row in $\mathbf{G}$ are minor when compared to calculating the gain matrix itself.

With this in mind, the state vector is chosen to be

$$\boldsymbol{x} = [\,\mathrm{OCS}, \mathrm{CO}, \mathrm{O}_3, \mathrm{CO}_2/\mathrm{N}_2\mathrm{O}, \mathrm{H}_2\mathrm{O}^1, \mathrm{H}_2\mathrm{O}^2, T_s, T^1, T^2, T^3, T^4]^{\mathrm{T}}, \tag{7}$$

where the superscript indexes the vertical location of the retrieved atmospheric layer, as visualised in Fig. 3, and the absence of a superscript for a gas implies a total column amount. Specifically, the natural logarithm of the volume mixing ratios (VMRs) is retrieved to enforce positivity in all of the gases and dampen the effect water vapour variability may have upon the results.

The term $T_s$ represents ground surface temperature. Emissivity is not included in the state vector because the emissivity Jacobian is highly spectrally correlated ($> 0.9$) with the surface temperature Jacobian and indistinguishable from the other without strict use of *a priori*. Therefore, considering the surface emission term in the equation of radiative transfer, it is clear that the retrieved quantity is effectively $\epsilon_s T_s$ for spectrally grey emissivity. It is important to note that spectrally changing surface emissivity across the range $2040 - 2080\,\mathrm{cm}^{-1}$ is currently not accounted for and may influence the OCS results over

land. However, spectral features of solids and liquids tend to be much broader than gases such that a grey approximation may be valid. Another source of error that may be more important than non-grey emissivity is the fact that all atmospheres in the ensemble were modelled with a surface emissivity of 0.99, which neglects downwelling radiation reflected back into the optical path. In both cases, observations over desert will be affected the most with minimal emissivity impact over water and dense vegetation.

The ratio of $\mathrm{CO}_2$ to $\mathrm{N}_2\mathrm{O}$ is included instead of the two separately to improve the conditioning of the inverse problem; which means that there is not enough independent information in the measurement to estimate both gases and atmospheric temperature without added constraints. Whilst $\mathrm{N}_2\mathrm{O}$ is a low variability gas that does not overlap with the OCS spectral features, the point of including it in the ratio is to account for variations in $\mathrm{CO}_2$ that may affect the OCS estimate. The downside to retrieving a ratio of two gases is that the knowledge of whether the numerator is enhanced versus a depletion of the denominator, and vice

versa, is sacrificed for the improved independence of elements in the state vector.

As shown in Fig. 3, four bulk layers of atmospheric temperature are retrieved ranging from the lower troposphere through the stratosphere. Additionally, two layers of water vapour are retrieved. The first layer is the lower-most troposphere that primarily accounts for water vapour continuum effects between absorption lines in the Jacobian as a result of self-broadening from $\mathrm{H}_2\mathrm{O} - \mathrm{H}_2\mathrm{O}$ collisions. The perturbation of the second layer peaks at $600\,\mathrm{hPa}$, but includes contributions from the remaining

upward levels of the atmosphere and accounts more for the absorption feature centres.

Since the statistical distributions of temperature and water vapour vertical profiles are well known, the resulting estimates can be constrained to scenarios found on Earth where clearly unphysical profiles are excluded with a negligible loss of sensitivity. Furthermore, since atmospheric temperature and water vapour are physical correlated, it is possible to represent this effect in the prior covariance. Thus, the 80 atmosphere ensemble was vertically binned down to the bulk layers of the retrieval and used

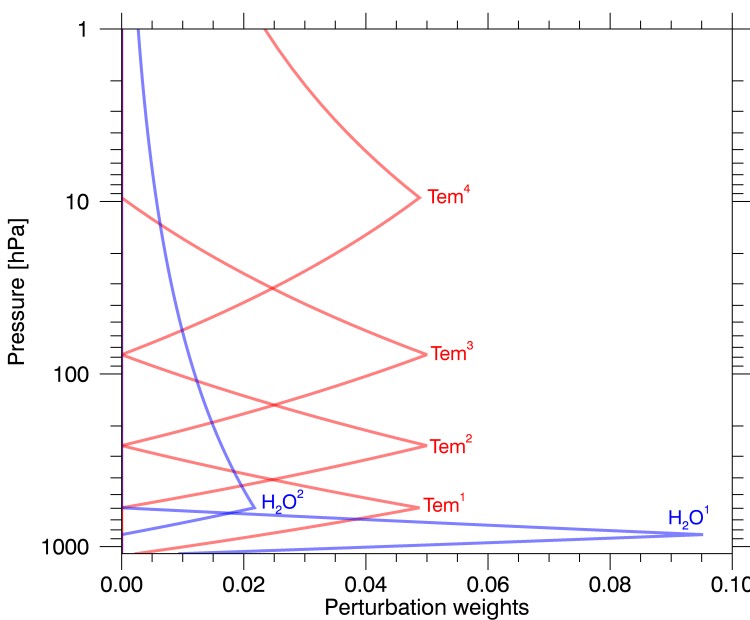

**Figure 3.** The vertical representations of the temperature and water vapour Jacobians are shown. These represent triangular pertubations as opposed to rectangular (even weighted) vertical perturbations.

to calculate the sample covariance matrix, which includes the cross-state physical correlation terms. The subsequent correlation matrix is shown in Fig. 4 with the standard deviations annotated along the diagonal elements. This sample covariance is then used as a sub-matrix ($6 \times 6$) within the prior covariance ($11 \times 11$) to constrain the water vapour and temperature portion of the retrieval to physical values within the global range. Further correlation terms between the remainder of the state elements are assigned to values of zero.

As a caveat, all elements of the state vector, including OCS, are technically constrained with finite values in the diagonal of the prior covariance. This is primarily for the purposes of developing a test-bed iterative retrieval that utilizes the Levenberg-Marquardt method, which will be discussed next. OCS variability in the prior covariance is assigned to be 200%. CO and $O_3$ variability is assigned 100%, the $CO_2/N_2O$ ratio is set to 10%, and surface temperature $20\,K$. However, this is such a weak constraint that the DFS for the OCS total column is close to one for all atmospheres and, therefore, effectively unconstrained.

### 3.4 Parameter validation using an iterative retrieval

Validation of the retrieval framework, as previously defined, is crucial towards developing confidence in the resulting estimates. Without analysing external data, one can show using an iterative retrieval that:

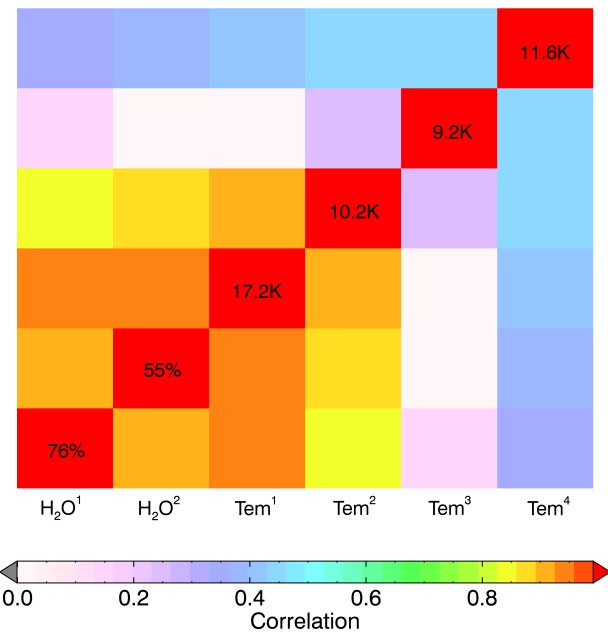

**Figure 4.** The correlation matrix is shown for the sample covariance of the $H_2O$ and atmospheric temperature layers calculated from the 80 atmosphere RTTOV ensemble. The standard deviations are annotated along the diagonal elements for reference.

1. The estimates converge during iteration.

2. The OCS spectral signature is noticeable in the residual spectrum of the converged result when excluded from the state vector and all other parameters are retrieved.

3. The variability of the converged residual spectrum over many pixels is similar to the expected instrument noise.

5   Each point is discussed in turn.

    The iterative retrieval was written as a test-bed for the faster linear scheme; so the spectral range, state vector, and prior covariance are the same as previously defined. This nonlinear approach is based on the Levenberg-Marquardt method as discussed in Rodgers (2000, ch. 5.7). The prior state is taken to be equal to the initial state, which is selected on a pixel by pixel basis from the ensemble of atmospheres. Each model atmospheric spectrum is compared against the measured spectrum

10  and the $j^{\text{th}}$ atmosphere that minimizes the spectral cost, i.e.,

$$\chi_j^2 = [\boldsymbol{y} - F(\boldsymbol{x}_j)]^{\text{T}} \mathbf{S}_\epsilon^{-1} [\boldsymbol{y} - F(\boldsymbol{x}_j)], \tag{8}$$

is chosen as the starting point. For atmosphere selection, only the diagonal of $\mathbf{S}_\epsilon$ is used to save computation time when rastering through the 80 atmospheres. Scenes with calculated cloud fractions from the Advanced Very High Resolution Radiometer (AVHRR) embedded data (Saunders, 1986) greater than 20% are not included. Based on this methodology it was found that the majority of IASI pixels converged on a result that reduced the $\chi^2$ cost function. While the presentation of the retrieval development to this point may appear overly streamlined or ad hoc, in reality this test for convergence can be used as a figure of merit and was repeated methodically numerous times as the state vector and prior covariance were modified until settling on the parameters defined in the previous section. The details of state vectors and prior covariances resulting in diverging iterations are not discussed for brevity.

OCS signatures can be shown in the converged residual spectrum (IASI minus RFM) if all other contributing parameters are retrieved. This is done by removing OCS from the state vector while retrieving the other 10 in its absence. Figure 5 shows an example of this for an IASI pixel in the North Atlantic off the coast of Iceland where the retrieved surface temperature is $281\,\mathrm{K}$. Notice that the OCS spectral signature is clearly above the IASI noise level for a particularly low surface VMR estimate of $404\,\mathrm{ppt}$ and matches well to the predicted OCS residual of the same VMR. It is important to keep in mind that Fig. 5 is for a single pixel without any spectral averaging to reduce instrument noise. Also apparent, is a substantial spike in the residual centred at $2077\,\mathrm{cm}^{-1}$. This feature is presumably due to line mixing errors within the RFM for the $CO_2$ Q-branch located at this position. Therefore, these particular channels should be avoided as they are poorly modelled.

Once all physical parameters that contribute to the signal above the noise level are accounted for through the joint retrieval, then the standard deviation of the spectral difference between the observation and the model, i.e., the residual, should be equal to the instrument noise. If this is not the case, then any parameters that are not completely accounted for will show an associated spectral feature in the standard deviation of the residual spectra. To test this posit, the iterative retrieval was run over 600 pixels in a $10° \times 10°$ latitude and longitude box in the Equatorial Pacific Ocean. The variability of the sample residuals is shown in Fig. 6 along with the average instrument noise profile in units of brightness temperature. Observe that the variability of the residuals matches closely to the average instrument noise with the exception of a few spectral features due to water vapour (Fig. 1). Therefore, the retrieval and associated state vector sufficiently account for the noticeable physical parameters aside from water vapour, which could be further resolved with more vertical levels along the profile.

There are three options to pursue with regards to unresolved, but influential, $H_2O$ levels in the retrieval. Firstly, this effect could be tolerated as an unaccounted source of error in the OCS estimates. Secondly, additional layers of $H_2O$ could be included in the state vector and jointly retrieved with an updated prior constraint. Thirdly, these associated $H_2O$ spectral features could be treated as effective noise within the measurement covariance, thus decreasing the sensitivity of the retrieval to variations in the water vapour vertical profile.

The first option is undesirable because there is clearly evidence supporting further treatment of $H_2O$. At first attempt, three layers of $H_2O$ were included in the state vector with a new prior covariance derived from the 80 atmosphere ensemble. However, it was found that this formulation did not converge unless a much stronger prior constraint was constructed. Therefore, these spectral variations for $H_2O$ were instead treated as noise by creating a vector of scaling factors that increased the diagonal of the measurement error covariance accordingly. This was accomplished by taking the ratio of the variance of the residual

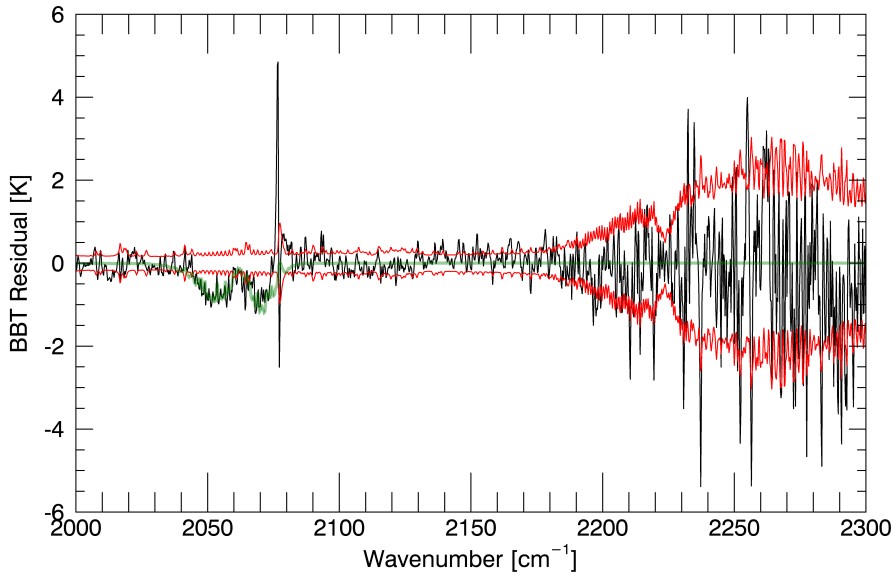

**Figure 5.** The residual spectrum between the IASI observation and the converged estimate from the RFM without modelling OCS is shown as the black line. The red line depicts the instrument noise level specific to the observed surface temperature of $281\,\mathrm{K}$ and atmospheric temperature profile. The green line represents the expected OCS signal in the residual for the retrieved VMR of $404\,\mathrm{ppt}$.

spectra over the square of the IASI instrument noise and setting any values less than one to unity. Thus, making the retrieval less sensitive to unretrieved layers of water vapour. All estimates of OCS from this point further include scaled variances for every diagonal element in the measurement error covariances.

### 3.5 Channel selection

Spectral channels in remote sensing tend to be highly correlated, not only by the gas specific rotational-vibrational energy transitions, but through other physical effects such as temperature and pressure. In other words, each channel does not normally add independent information and contains a certain amount of redundancy. In theory, adding more channels to the estimate always increases the total information content to varying degrees. In practice, there are spectral channels that contain more information than others such that adding channels of negligible importance does little to improve figures of merit (like DFS and posterior uncertainty), but increases sensitivity to unaccounted physical parameter errors. One method to improve the robustness of a retrieval by reducing sensitivity to unaccounted parameters is to select a subset of spectral channels that contains the majority of information while excluding the remaining channels that negligibly contribute.

Channel selection was performed over the $2000 - 2300\,\mathrm{cm}^{-1}$ range in order to remove these spectral channels of little importance. One option is to remove channels while maximizing a figure of merit for the joint retrieval as a whole. Another

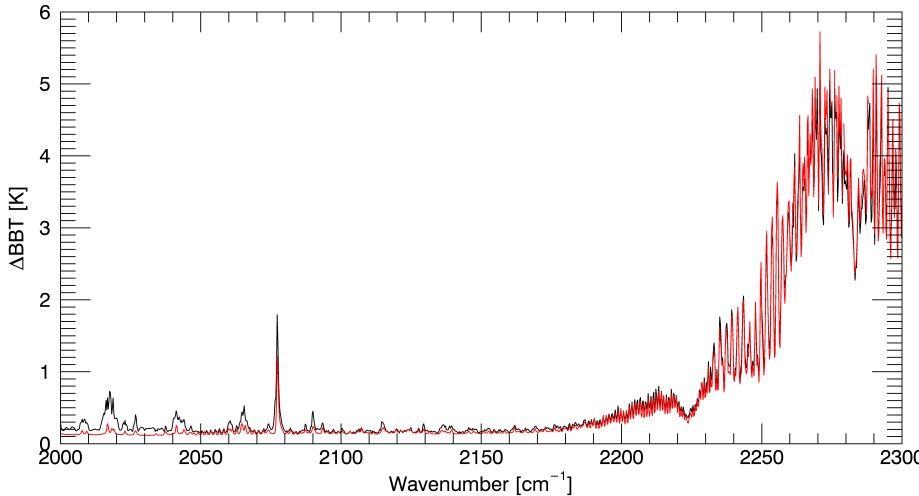

**Figure 6.** Black: The sample standard deviation of the residual spectra between the IASI measurements and the converged model spectra for an ensemble of 600 pixels from the tropical Southern Pacific Ocean. Red: The average instrument noise (NE$\Delta$T) for the IASI observations.

is to maximize just the OCS portion of the retrieval at the expense of the other retrieved parameters. Since the other states are included just to improve the OCS estimates, the latter option is chosen here.

OCS is so weakly constrained that attempting to maximise the DFS is not appropriate in this instance. In the unconstrained case, the DFS is not defined for maximum likelihood estimates. However, it is always desirable to minimize the posterior uncertainty, whether constrained or not. In this case, just the uncertainty component of OCS is considered:

$$\hat{\sigma}_{\text{OCS}}^2 = \hat{\mathbf{S}}_{1,1}^x, \tag{9}$$

where $\hat{\mathbf{S}}_x$ is defined in Eq. (3) and the subscript index denotes the first diagonal element corresponding to OCS.

The selection begins by first finding the best two spectral channels that minimise $\hat{\sigma}_{\text{OCS}}^2$ after calculating all possible two channel combinations. Then a third channel is selected by adding all remaining channels individually and choosing the one which reduced $\hat{\sigma}_{\text{OCS}}^2$ the most. This process is repeated until all spectral channels have been ranked according to their contribution towards minimizing the posterior uncertainty of OCS. The resulting channel ranking for a mid-latitude atmosphere is visualised in Fig. 7 where the best two channels estimate OCS uncertainty to be nearly 50% while including all 1201 channels reduce the uncertainty to just over 10%. Notice that the first 20 channels reduce uncertainty by a factor of two from the initial pair, but it takes the remaining channels to gain another factor of two reduction. For this retrieval the top 100 channels were retained, which yield an uncertainty of just 12% (versus 10%) for this particular atmosphere with 12 times fewer channels.

The resulting selected channels are shown in Fig. 8 for reference. Channels are selected from this method covering the entire spectral range, rather than just the $40 \, \text{cm}^{-1}$ OCS interval, because these outside channels contribute to the other 10 parameters jointly estimated that help improve the OCS retrieval. Channels are only selected in so far as they contribute to better

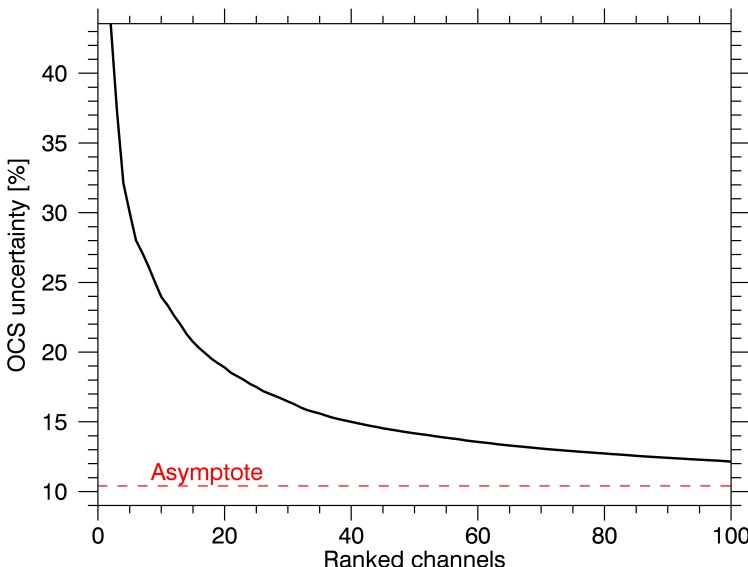

**Figure 7.** Ranked spectral channels are shown for a mid-latitude atmosphere based on their contribution towards minimizing the posterior uncertainty of OCS. The asymptote from including all 1201 channels is shown as the dotted red line.

OCS estimates. The $CO_2$ Q-branch at $2077\,cm^{-1}$ was avoided by heavily penalizing these channels within the measurement covariance prior to running the selection. Notice that the selected channels largely avoid the majority of $H_2O$ absorption features and frequently select the between band channels associated with water vapour continuum.

### 3.6 Selecting the initial atmosphere

The validity of the linear retrieval is contingent upon the choice of initial atmosphere. The initialisation point should be sufficiently close enough to the observed atmosphere that a single step places the estimate within the uncertainty level of the true state being observed. Failure to do so results in retrieval error due to the nonlinearity of the formulated problem. So how should an initial atmosphere be selected in order to minimize the nonlinearity error? Three possible techniques are analysed for determining the initial atmosphere that do not require rerunning the forward model.

1. Select the initial atmosphere whose model spectrum minimizes the spectral cost function in Eq. (8), as previously discussed. This method essentially picks the atmosphere whose mean spectrum (i.e., averaged along the spectral axis) is closest to the IASI observation for the selected spectral channels. The diagonal may be used to approximate $S_\epsilon$ to speed up the process of running through the 80 atmospheres for each pixel since the selected channels contain few adjacent pairs.

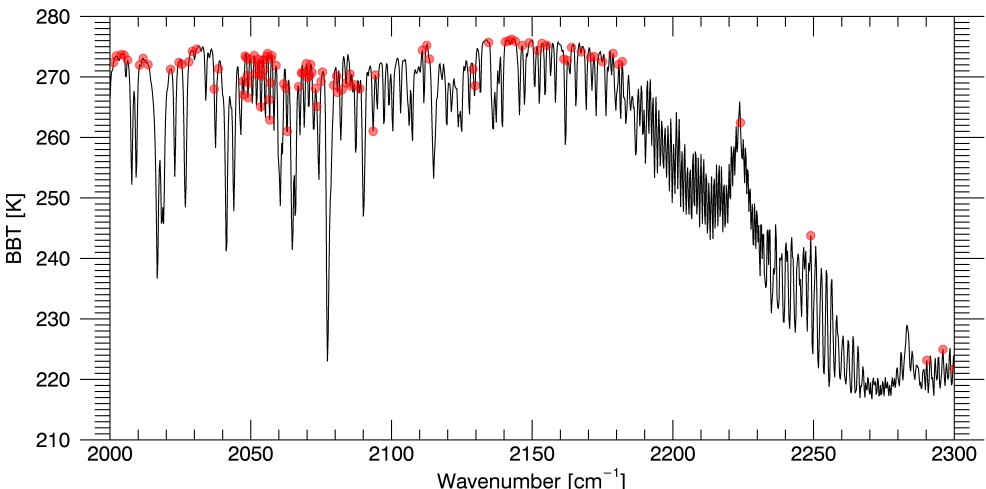

**Figure 8.** The top 100 spectral points (red circles) ranked in Fig. 7 from the channel selection are shown for reference as compared to all 1201 channels from the observed spectrum (black line) for a mid-latitude atmospheric scenario.

2. Another method is to estimate what the model spectrum would be after the retrieval, within the linear framework of the problem, and then select the atmosphere which minimizes the projected spectral cost. The retrieved state can be linearly projected back into spectral space to estimate the posterior spectrum,

$$\hat{\boldsymbol{y}}_j = \mathbf{K}_j \left(\hat{\boldsymbol{x}}_j - \boldsymbol{x}_j\right) + F(\boldsymbol{x}_j). \tag{10}$$

If $\hat{\boldsymbol{y}}$ is used instead of $F(\boldsymbol{x})$ in Eq. (8),

$$\chi^2_{pr;j} = \left[\boldsymbol{y} - \mathbf{K}_j \left(\hat{\boldsymbol{x}}_j - \boldsymbol{x}_j\right) - F(\boldsymbol{x}_j)\right]^{\mathrm{T}} \mathbf{S}_\epsilon^{-1} \left[\boldsymbol{y} - \mathbf{K}_j \left(\hat{\boldsymbol{x}}_j - \boldsymbol{x}_j\right) - F(\boldsymbol{x}_j)\right], \tag{11}$$

and $\hat{\boldsymbol{x}}$ is expanded using Eq. (2), then the resulting projected cost is given by

$$\chi^2_{pr;j} = \left[\boldsymbol{y} - F(\boldsymbol{x}_j)\right]^{\mathrm{T}} \left(\mathbf{K}_j \mathbf{G}_j - \mathbf{I}\right)^{\mathrm{T}} \mathbf{S}_\epsilon^{-1} \left(\mathbf{K}_j \mathbf{G}_j - \mathbf{I}\right) \left[\boldsymbol{y} - F(\boldsymbol{x}_j)\right]. \tag{12}$$

It is important to note that $\mathbf{K}\mathbf{G}$, unlike $\mathbf{G}\mathbf{K}$, is generally not equal to the identity matrix in the unconstrained least-squares retrieval.

3. Finally, the third method considered is to train a vector operator to predict the non-linear error in OCS based upon the spectral difference between the initial model and measurement spectra. To do this, all possible permutations ($80 \times 79 = 6320$) of using one state from the 80 atmosphere ensemble as the initial point to retrieve another atmosphere from the ensemble are calculated to yield two matrices; an array of initial spectral differences ($\Delta\mathbf{BBT}$ of size $6320 \times m$) and a

vector of corresponding linearly retrieved OCS errors ($\delta\mathbf{OCS}$ of size $6320 \times 1$). The goal is to determine a prediction vector ($\boldsymbol{a}$ of size $m \times 1$) that approximates the following equation:

$$\delta\mathbf{OCS} = \Delta\mathbf{BBT} \times \boldsymbol{a}. \tag{13}$$

However, since there are only 80 independent atmospheres considered, Eq. (13) is actually underdetermined rather than overdetermined as it may appear at first glance. Therefore, the dimensionality of the problem must be reduced if Eq. (13) is to be successfully inverted to find $\boldsymbol{a}$. Subsequently, $\Delta\mathbf{BBT}$ is decomposed into singular vector components, $\Delta\mathbf{BBT} = \mathbf{U}\Lambda\mathbf{V}^{\mathrm{T}}$, where $\mathbf{U}$ and $\mathbf{V}$ are the left and right singular vectors, respectively, and $\Lambda$ is a diagonal matrix of its singular values. The inner dimensions of $\mathbf{U}$ and $\mathbf{V}^{\mathrm{T}}$ are then ranked in order of decreasing singular values and truncated at 79. Equation (13) is then recast as

$$\mathbf{U}^{\mathrm{T}}\delta\mathbf{OCS} = \mathbf{U}^{\mathrm{T}}\left(\Delta\mathbf{BBT}\right)\mathbf{V} \times \boldsymbol{a}', \tag{14}$$

where the truncated least-squares solution to $\boldsymbol{a}'$ is calculated. Finally, the prediction vector is found to be $\boldsymbol{a} = \mathbf{V}\boldsymbol{a}'$.

A fourth possible method would be to select an initial atmosphere based on the time of year and proximity to the observed pixel location. However, the RTTOV ensemble is not well suited for this particular selection method as the atmospheres were chosen to maintain statistical properties of a much larger ensemble and, therefore, are irregularly spaced in location and season. A separate ensemble of atmospheres parsed in regularly spaced latitude and longitude grids at monthly increments would be more appropriate. Therefore, this study excludes this fourth possible selection method.

The three listed initial atmosphere selection methods are compared using the RTTOV ensemble in the absence of instrument noise and contaminating parameters so that the error due solely to non-linearity is assessed. Each atmosphere of the 80 is used as a test case where the objective is to select an initial atmosphere from the remaining 79 which minimizes the error in the estimate while knowing the true OCS model value. In the ensemble all atmospheres contain the same OCS profile, because of the lack of information about its distribution and variability. The model OCS profile is $590\,\mathrm{ppt}$ at the surface, constant up to the tropopause, and steadily decreasing with altitude through the stratosphere. Therefore, the best initial atmosphere yields a retrieval step closest to zero, because OCS is a flat field throughout the model atmospheres.

Figure 9 shows histograms of the linear assumption error for the three discussed selection methods. Furthermore, a method of randomly selecting the initial atmosphere was also analysed to provide a baseline for comparison. Evidently, the method that most frequently yielded retrieval errors near zero was from selecting the atmosphere that minimized the projected cost (method 2). This was followed by matching the mean spectrum (method 1). Predicting the retrieval error (method 3) worked in the sense that it outperformed the baseline of random selection, but provides larger error than the other two methods.

One may conclude that the initial atmosphere should be selected based upon minimizing the linearly projected cost. However, this was attempted with real data and in practice it became clear that a grossly non-linear starting point, such as using an initial polar atmosphere for an observation in the tropics, may occasionally be projected to outperform all other atmospheres. This is because linear analysis is only valid in the nearly linear to moderately linear regimes. Therefore, the method of selecting an

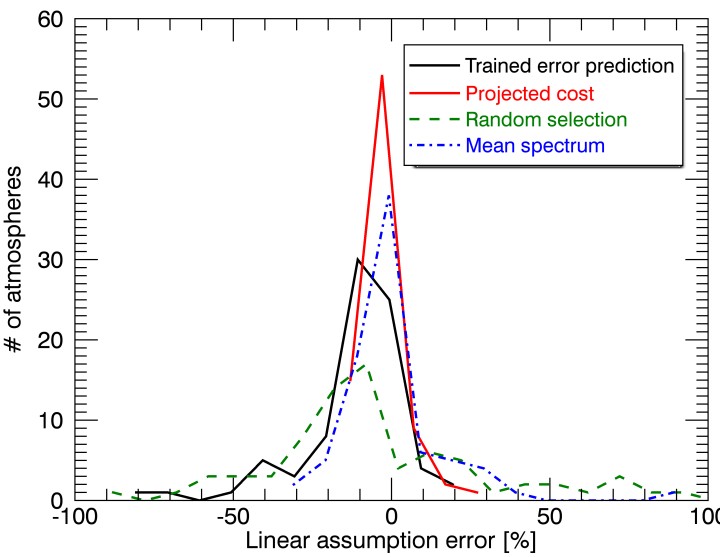

**Figure 9.** Histograms of the OCS retrieval error due to non-linearities are shown for four methods of selecting the initial atmosphere, $x_0$. The first enumerated method in this section is labeled 'mean spectrum', the second 'projected cost', and the third 'trained error prediction'.

initial atmosphere by minimizing the difference in the mean spectra, Eq. (8), was used in the work presented here because it avoids this particular problem.

The expected non-linearity error is given by the width of the distributions in Fig. 9 and the mean spectrum method was found to have an error of 11% on average. This analysis was also performed using all spectral channels, i.e., without channel selection, and found to yield an average error of 19% for this method. Thus, channel selection is crucial towards improving the OCS retrieval, because it makes the problem almost twice as linear. At first this may seem counter-intuitive, because reducing the number of spectral channels results in reduced linear assumption error. From an information perspective, adding ranked spectral channels always increases information content until the addition becomes asymptotically negligible (see Fig. 7). However, in an imperfect retrieval, adding spectral channels of minor importance provide additional inputs for systematic errors to propagate into the estimate. Therefore, channel selection is a technique to reduce the effect of systematic errors, such as neglecting non-linearity.

### 3.7   Geographical considerations

Lower most tropospheric pressure is influential in the OCS retrieval not just through the direct effect of pressure broadening the spectral features near the surface, but also because of pressure dependent water vapour continuum effects in the lower troposphere that overlap with all OCS spectral lines. Surface pressure variations due to geographical altitude must, therefore, be

accounted for in some way. If not, then the Jacobians and initial column amounts will misrepresent the observation, especially over mountain ranges and high plateaus.

To do this, separate atmospheric ensembles of model spectra, gain matrices, and initial values were created for surface pressure scenarios of 1030, 900, 800, and $700\,\mathrm{hPa}$. Average surface pressure was tabulated from ECMWF reanalysis data and stored as a reference field. Prior to computing the linear estimates of OCS, the surface pressure for each IASI pixel based on its latitude and longitude is interpolated from the saved map. Then the appropriate ensemble is selected based upon whether the interpolated surface pressure falls outside or within the bounds of 950, 850, or $750\,\mathrm{hPa}$. Applying this method noticeably removed any high terrain artefacts that systematically appeared in the OCS estimates.

Additionally, temperature contrast between the ground surface and lowest atmospheric layer affects the sensitivity of the OCS estimates, as shown in Fig. 2. Thermal contrast is a particular problem over land, and especially deserts in the summer, where the surface is heated by solar absorption to values occasionally greater than $15\,\mathrm{K}$ above the atmospheric surface temperature. In the deep Antarctic, there can be a negative thermal contrast where the surface is actually colder than the atmosphere and absorption lines switch to emission features. This effect is far less important over the oceans, because the heat capacity of water is so great that thermal contrast tends to be slightly positive with less variability.

Therefore, the method employed in this work is to treat IASI observations over ocean as having a routine thermal contrast of $+3\,\mathrm{K}$, while allowing for greater variation over land. Instead of selecting from 80 atmospheres over land with one thermal contrast option, the ensemble is grown to include scenarios of -5, 3, 10, and $15\,\mathrm{K}$ of thermal contrast. So an observation over land has 320 possible atmospheric initialization points to select from. As previously mentioned, the model atmospheric spectrum that most closely matches the observed spectrum determines which atmosphere is selected as the initial point.

## 3.8 Quality filtering

In an iterative retrieval, high confidence in the estimate is obtained by verifying that the retrieval converged on a minimum $\chi^2$ value. This may not be the correct minimum, but the fact that a minimum was found suggests that the framework of the problem is behaving in a consistent way. In a one-step linear retrieval the forward model is not recalculated for each individual pixel in order to save computation time. Incidentally, other metrics of quality must be evaluated in order to identify and exclude retrievals that have likely gone awry. The steps to filter the OCS estimates for quality are described in detail.

First, any IASI pixels with an AVHRR cloud fraction of 20% or greater are excluded from consideration prior to computing the retrieval. The presence of cloud introduces highly non-linear behaviour that must be modelled properly if the OCS estimates are to be trusted. This AVHRR cloud fraction product is not perfect and routinely flags sea ice as cloud. However, the vast majority of the time it provides a robust and accurate estimate of the amount of cloud filling the IASI pixel. Therefore, cloudy scenes are simply avoided in favour of clear sky observations.

Next, viewing angles noticeably affected by sun glint are excluded from the retrieval by calculating the specular solar reflection angle based upon the solar and satellite zenith and azimuth angles (Vincent, 2016, Ch. 2.3) and removing pixels where this angle is less than $18°$. Additionally, there is a slight overestimation of OCS when observing towards the limb. Rather than attempting to parametrise or mitigate this effect, observations with an air mass factor relative to nadir greater than

1.47 are avoided. This removes the very far edges of the IASI scan where the surface zenith angle is greater than $47°$. For surface zenith angles less than this value, limb effects were not noticeable. Fortunately, the overlap of IASI-A with IASI-B is greater than this angular width, so no spatial gaps in coverage are introduced as a result.

Since the retrieval jointly estimates other physical parameters in conjunction with OCS, there is further opportunity for common sense filtering for quality. For example, if the retrieved surface temperature falls outside of the range between $230 - 340\,\mathrm{K}$, then that pixel is removed from consideration. Furthermore, if the lowest level of retrieved water vapour has a VMR greater than 4%, then the observation is clearly not represented properly and those OCS estimates are excluded.

Finally, the projected spectral cost from Eq. (12) can be used as a retrieval diagnostic given the fact that the atmosphere with the smallest initial spectral cost was selected as the initialization point. The expectation value of the projected cost should be approximately equal to the number of spectral channels ($m = 100$) if the retrieval were ideally linear. Since the problem is not linear, the average projected cost will certainly be greater than $m$. However, the magnitude of the projected cost provides a useful prediction as to how well the retrieval may perform. Thus, a reasonable criteria for accepting a retrieved pixel is given by

$$\frac{\chi^2_{pr}}{m} < 2. \tag{15}$$

Aside from filtering against cloudy scenes, this provides the strictest quality test of those mentioned and highlights geographical areas that are poorly represented by the modelled atmospheric ensemble.

## 4   OCS results from 2014

The entirety of IASI-A and B data from 2014 ($19.4\,\mathrm{Tbytes}$) was downloaded and processed in this study using the previously described linear retrieval technique. OCS total column median values are shown in Figs. 10–15 for two month intervals in latitude-longitude bins of $0.5° \times 0.5°$. Median OCS values combined from all data in 2014 are shown in Fig. 16. The median was chosen instead of the mean, because the retrieval actually estimates the logarithm of the total column to enforce positivity and when raised to the exponential introduces positive skewness into the distribution of estimates. In other words, the spread of OCS estimates does not follow a Gaussian (normal) distribution and the heavy tail towards overestimation is mitigated by taking the median rather than the mean. The median also dampens the effect anomalous cases have upon the statistics of the distribution, whereas one bad pixel resulting in a wildly high or low total column amount could artificially dominate the mean.

The number of pixels per bin passing the quality and cloud free criteria is also shown for reference. Only bins containing three or more observations are shown and any areas with two or less observations are considered missing and coloured grey. Areas that are systematically low in number of observations are either routinely flagged as cloudy or routinely predicted via the projected cost to poorly model the observation. Notice that areas of sea ice towards the poles are consistently absent, which is due to AVHRR cloud flagging. However, persistent glaciers over land contain many more observations and do not experience this false-positive cloud flagging effect. Alternatively, desert areas during the day-time in local summer are frequently cloud free and marked as such, but routinely fail the quality check and contain few estimates. This signifies that the model atmospheres

in the ensemble fail to closely match summer desert scenarios that are sun illuminated, perhaps because of lower surface emissivity that increases solar and downwelling reflections that are currently not modelled.

The sample standard deviation of OCS per spatial bin over the two month period is shown in the bottom row of Figs. 10–15. This gives an estimate of the width of the OCS distribution based upon the sampling of retrieved values. In an iterative retrieval, the posterior uncertainty from $\hat{\mathbf{S}}_x$ is normally used to represent the error of the retrieval. However, within the linear framework of this method the posterior uncertainty derived from the initial guess will systematically underestimate the true error of the OCS retrieval. Thus, the sample standard deviation provides a metric that is a combination in quadrature of retrieval noise, natural OCS variability, and errors due to unaccounted parameters. At a minimum, the sample standard deviation of OCS will be no less than the retrieval noise, assuming there is a sufficient number of samples. Areas that are clearly dominated by retrieval noise are the Antarctic plateaus, Greenland, and high latitude land in the Northern Hemisphere during winter.

## 4.1 Estimates over ocean

Beginning with the oceans, there is a clear correspondence of OCS estimates observed between day and night. Prior to filtering based on the solar reflection angle, it was apparent that sun glint was an issue for estimates over water, especially near the equator. However, by excluding observations along the specular path this issue was mitigated such that the day and night estimates resemble each other. This is the expected result because variations in thermal contrast from the day to night over water should be fairly small. Therefore, OCS should be equally detectable over water regardless of the time of day.

OCS estimates throughout the year show that there is a consistent feature of elevated OCS in the South Pacific off the coast of South America between 0 and $-30°$ latitude (Fig. 16, point 1) that matches well to the direct OCS emissions modelled in Launois et al. (2015b). While there is some variation throughout the year, this particular feature remains relatively constant regardless of season. In contrast, further South there is a large OCS signal that appears to align with the Antarctic Circumpolar Current (ACC) in both day and night observations (Fig. 16, point 2). This particular feature shows a large seasonal variation with maxima occurring during southern hemisphere summer and minima during winter when incident solar radiance is low. This is the seasonal cycle one would predict if the primary source of OCS were photochemically reduced CDOM. Before too much is concluded, it is important to acknowledge the fact that this OCS signal at $-60°$ latitude may be a false positive resulting from a temperature artefact specific to the ACC. However, it is certainly worth further investigation.

Northern hemisphere ocean areas appear to have maximum OCS signal between March and June (local spring) with minimum values as the season approaches winter. Once again this is consistent with how the incident solar radiation varies with season for photochemical production. OCS features that particularly stand out in these areas are the tropical enhancement during May to June coming off the coast of Baja California (Fig. 12, point 3) and the high latitude structures south of Greenland and north-east of Iceland (Fig. 12, point 4) in this same time period. Additionally, there appears to be a consistent enhancement of OCS in the northern Indian Ocean by the Saudi Arabian Peninsula (Fig. 13, point 5), which also resembles the model in Launois et al. (2015b).

Interestingly, there is an OCS feature over the Pacific Ocean between Japan and Alaska (Fig. 11, point 6) that is in phase, but one month delayed, with the high OCS signal over the east of China and the Tibetan Plateau. This ocean feature begins in

January and February, reaches maximum in March and April, and then dissipates by August. Whereas the OCS land signal over China grows substantially in November and December and then is closer to background levels in May and June. One possibility is that the enhancement over the ocean between Japan and Alaska is an OCS plume originating from China transported by the easterly zonal winds that dissipates when OH concentrations increase during spring and summer. On the other hand, the two

signals may be purely coincidental and indicative of two unrelated sources of OCS or other atmospheric characteristics that produce artificially high estimates over these regions.

## 4.2 Estimates over land

Satellite retrievals over land are subject to a greater number of surface type variations than over ocean. As a result, there are more variants contributing to the signal that may require a modelled response; such as emissivity, altitude, surface facets,

reflectance distribution functions, and snow cover. Therefore, one must analyse spatially sharp OCS gradients over land coinciding with geographical features and overly distinct land-sea boundaries with a certain amount of scepticism.

Recent work by Glatthor et al. (2015) and Berry et al. (2013) have shown that there should be a noticeable depletion of OCS over the Amazon and Congo rainforest areas due to strong vegetative uptake. This is indeed what is observed in these data, especially for the observations made during the day (Fig. 16, point 7). The Amazon and Congo areas show OCS total

columns approximately 10–20% less than what is estimated over nearby oceans at the same latitude. Therefore, these results are consistent with the idea that vegetative uptake is a significant sink of OCS. However, notice that the night-time estimates (Fig. 16, point 8) tend to be slightly greater than the day-time, which may be indicative of a physical OCS process with a diurnal signal. It is also possible that this effect is an artefact of the retrieval. One may quickly blame thermal contrast between day and night observations; except that it is the wrong way around from what is expected. For example, areas over desert like the

Sahara and much of Australia show low OCS at night and higher OCS during this day. This is because solar heating increases thermal contrast, which makes trace gases more detectable. During the night, these low humidity areas quickly radiate away their heat and come closer to thermal equilibrium between the surface and the lower atmosphere, thus decreasing sensitivity to OCS. Therefore, if the higher night-time OCS signal over the rainforests is not physical, then it is unlikely to be solely due to thermal contrast.

Along this same vein, notice that the high latitude areas over land near the Arctic (Fig. 10, point 9) show substantially less OCS in the Winter months than any other time of year. The high standard deviations of the estimates during this time show that these low OCS values may be due to a loss of detectability as the signal drops from cold temperatures over land. The same can be said for estimates over Greenland and most of Antarctica throughout the entire year, i.e., the SNR of OCS is too low to have much confidence in retrievals over these areas. However, sensitivity appears to return for estimates over northern Canada

and Russia during spring, summer, and fall.

Additionally, there are several areas over land where there are particularly high OCS signals. Much of the continental United States show OCS estimates greater than ocean values at similar latitudes. The United States OCS signal appears to be maximum during March to April and minimum during July and August with a slow build up back to March. If these estimates

are indicative of the true OCS levels, then the July to August minimum coincides with peak vegetative uptake for regions at this latitude. Sources of OCS in the United States, especially anthropogenic and biomass burning, are currently poorly understood.

Many regions in the Middle East and the north African Mediterranean coast also show very specific enhancements of OCS estimates. It is possible that there exists a surface emissivity feature in these regions that routinely yields spurious elevated OCS values. However, some of this effect is likely mitigated by the process of calculating the projected cost of the retrieval and removing pixels where the model initial atmospheres are predicted to poorly represent the scene. Therefore, it may also be possible that these signals are real and there are large sources of OCS creating local enhancements. If this signal represents physical OCS amounts, then the source is more likely to be anthropogenic in nature given that the detail closely follows geographical boundaries of human population.

Finally, the areas of high OCS signal over China and the former Soviet republics east of the Caspian Sea especially stand out in displayed estimates. These are also areas of known $SO_2$ emissions due to industrial processes and energy production that are routinely modelled in chemical transport models, such as TOMCAT (Spracklen et al., 2005). While it is energetically unfavourable for $SO_2$ to convert to OCS, the two may be positively correlated in many physical situations, especially in anthropogenic processes that do not have strict methods in place to reduce $SO_2$ emissions.

## 5  Comparisons to NOAA flask samples

Total column estimates of OCS from the linear retrieval were also compared to VMR flask measurements of OCS collected by NOAA (Montzka et al., 2007). Although IASI total columns are different from localized point samples, the intent is to compare seasonal cycles to see if the two are temporally correlated. The Earth System Research Laboratory of NOAA collects surface air samples by flask from network sites located across the world to measure seasonal trends of numerous trace gases, including OCS. Flask measurements of OCS tend to have uncertainties within the range of $0.1 - 6\,\mathrm{ppt}$ and are normally sampled on a weekly basis, but may occur less frequently depending upon location. Further information and the OCS flask data themselves are found on-line at http://www.esrl.noaa.gov/gmd/hats/gases/OCS.html.

Figure 17 shows the seasonal trend comparisons for IASI total columns against the NOAA flask measurements for seven sample sites; four in the Northern Hemisphere and three in the Southern Hemisphere. The results are displayed in monthly increments throughout 2014 where the IASI retrieved total columns are binned within a $2°$ radius about the location of the NOAA site. The site abbreviations along with their latitudes and longitudes are shown in the plot titles. Additionally, the monotonic (Spearman's) correlation coefficient ($R$) and its associated p-values ($p_{val}$) are also displayed. Note that p-values represent statistical significance on a scale of $0 - 1$ where values close to zero show significance in the sampled correlation, while higher values fail to reject the null hypothesis. As a rule of thumb, p-values less than $0.05$ are generally regarded as statistically significant (Hung et al., 1997).

Comparing pressure specific VMR to total column amount can be tenuous if the true shape of the vertical profile differs greatly from the referenced profile. Furthermore, the flask samples are not exactly coincident with the IASI observations in space and time, so this combines to introduce a certain level of natural error that is difficult to isolate and quantify. However, by

analysing on a monthly basis, these effects may be mitigated where the desired outcome is to show correlation and consistency between the seasonal signals of the two.

Of the seven, the Harvard Forest (HFM) site shows the greatest correlation at $R = 0.88$. It is important to point out that the flask samples here are taken immediately above the forest canopy at $30\,\mathrm{m}$, while the IASI observations are most sensitive at mid-troposphere. Notice that the OCS flask VMR closely follows the total column trend during the winter months, but then drops proportionately much lower from June to September. Work discussed in Sect. 2.1 suggests that forests are strong sinks of OCS and, therefore, most active during peak summer-time photosynthesis. Therefore, one would expect this sort of surface drop at canopy level compared to the total column of OCS.

Perhaps the most important comparison is the Mauna Loa Observatory (MLO), because the air is sampled closer to the peak sensitivity of IASI at an altitude of $3.5\,\mathrm{km}$. Both flask VMRs and total columns show a clear seasonal cycle of OCS reaching maximum in late spring and minimum in early winter with a correlation coefficient of $0.76$. A similar comparison was made for the OCS retrievals using TES (Kuai et al., 2014) to NOAA flask measurements over Mauna Loa during 2011. They found a slightly higher correlation coefficient of $0.80$ for their seasonal analysis, which is expected given that the TES retrieval accounts for non-linearities by iteratively minimizing the joint cost function.

Correlations similar to Mauna Loa are found at Trinidad Head (THD), Cape Grim Observatory (CGO), and Palmer Station in Antarctica (PSA). However, the site at Mace Head (MHD) shows a lower correlation of only $0.54$ between the surface VMRs and the total columns. Inspection of both indicates that the OCS values at Mace Head are quite variable throughout the year with no clear seasonal behaviour. In this case, coincidence between flask samples and IASI observations becomes much more important due to the variable nature of OCS at this specific location on the west coast of Ireland.

Finally, the NOAA site located in American Samoa (SMO) actually shows a negative correlation between flask samples and IASI estimates. This is entirely due to the first two months of the year, January and February, while the remainder of the year shows a positive correlation. This early year depletion in the total column estimates can be visualized in Fig. 10. Notice that there is a spatial low in OCS total column that extends from the Indonesian islands well into the middle-south Pacific during this time of year. Since this is peak season for photosynthesis in the Southern Hemisphere, it is possible that American Samoa is downwind of Indonesia and northern Australia, strong OCS sinks for January and February, while the ocean surface near American Samoa is emitting OCS or its precursor gases. On the other hand, it is possible this feature is an artefact of some unsensed physical parameter or a weather effect yielding a nonlinear error biased consistently low.

## 6   Conclusions

A novel linear retrieval method was developed and applied towards making timely estimates of OCS total columns for the entirety of IASI observations from 2014. There are two components that make this retrieval scheme unique in comparison to current linear methods. First, physical parameters that influence the spectral observations over the wavenumber range used for OCS are directly accounted for by jointly retrieving them along with OCS. This differs from previous methods in that they tend to use an effective measurement covariance that treats the physical parameters not directly retrieved as noise. Second, an initial

linearisation point is selected from a global ensemble of atmospheres based on minimizing the spectral difference between the IASI and the modelled spectral radiances. This step is intended to make the retrieval more linear, thus reducing the need for iterative steps that rerun the forward model several times per pixel.

Additionally, an iterative retrieval for OCS was used as a test-bed to develop and validate the framework of the retrieval; i.e., the state vector, prior constraints, and initial atmosphere selection. Once this was accomplished, an ensemble of IASI observations over the Pacific Ocean was used to quantify the mean spectral residual for the converged estimates and showed that the majority of spectral channels match to within instrument noise, except the stronger water absorption features. Water vapour channels were then treated as noise by modifying the measurement covariance diagonals accordingly based on the mean spectral residual. Finally, channel selection was performed based on the OCS posterior uncertainty, reducing the number of channels from 1201 to 100, which ultimately made the OCS retrieval almost twice as linear.

The OCS estimates visualized in two month increments display many interesting features consistent with prior knowledge of its sources and sinks. For example, the day-time total columns show depletions in the OCS signal over tropical rainforests, which is consistent with the idea that vegetation is the strongest sink of OCS. The Pacific Ocean displays spatial features of elevated OCS that vary seasonally and appear to match the prediction made by Berry et al. (2013) that there is a large source in the Pacific Ocean, especially in the southern hemisphere. Interestingly, there is a clear band of high OCS estimates following the circumpolar current north of Antarctica, which is well known for consistent upwelling sustained by turbulent gyres. Additionally, regions of land showing high OCS estimates were found over China, the area east of the Caspian Sea, and northern coastal Africa leading to the Middle East. It is possible these land regions are emitting anthropogenic OCS or that there is some surface property unaccounted for that consistently leads to elevated estimates.

To validate the linear retrieval on a monthly basis, these OCS results were compared to surface VMR samples collected via flask by NOAA stations across the globe. It was found that five (three northern and two southern hemisphere) NOAA sites out of seven had seasonal cycle correlation coefficients greater than 0.7. Further comparisons to aircraft campaigns and zenith-viewing surface estimates of OCS may be attempted in the future.

In the absence of a large computational cluster, iteratively analysing forward models of radiative transfer may still be too time consuming to evaluate IASI data beyond individual and area specific events. In this case, one can reduce the accuracy of the retrieval by treating the problem within the linear framework presented in this paper while speeding up the computational process by a factor of roughly $10^4$ (depending upon the specific retrieval). Analysis of model scenarios suggests that the error due to ignoring non-linearities is about 11% globally for OCS. Since these linear estimates can be generated so rapidly, it is possible to use a monthly median or mean of linear OCS fields as the initial point ($x_0$) or even the *a priori* ($x_a$) to improve the efficiency and data quality of a constrained iterative retrieval.

Work presented here paid particular attention to OCS as an interesting test case. However, it is important to note that the linear retrieval method presented, of using a multi-element state vector to jointly account for other physical parameters and selecting an initialisation point from an atmospheric ensemble, can be applied to any trace gas for any nadir viewing instrument similar to IASI. While the OCS results require further validation, the presented OCS spatial fields are intriguing and may lead

to future understanding of its sources and sinks. Furthermore, this method can potentially provide additional insights for minor trace gases that are, as of yet, poorly quantified.

*Acknowledgements.* Portions of this work were funded by the United Kingdom's National Centre for Earth Observation and the United States Air Force. The views expressed in this article are those of the author and do not reflect the official policy or position of the United States Air Force, Department of Defense or the US Government. Additionally, we thank Steve Montzka of NOAA for permission to use the OCS flask data.

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

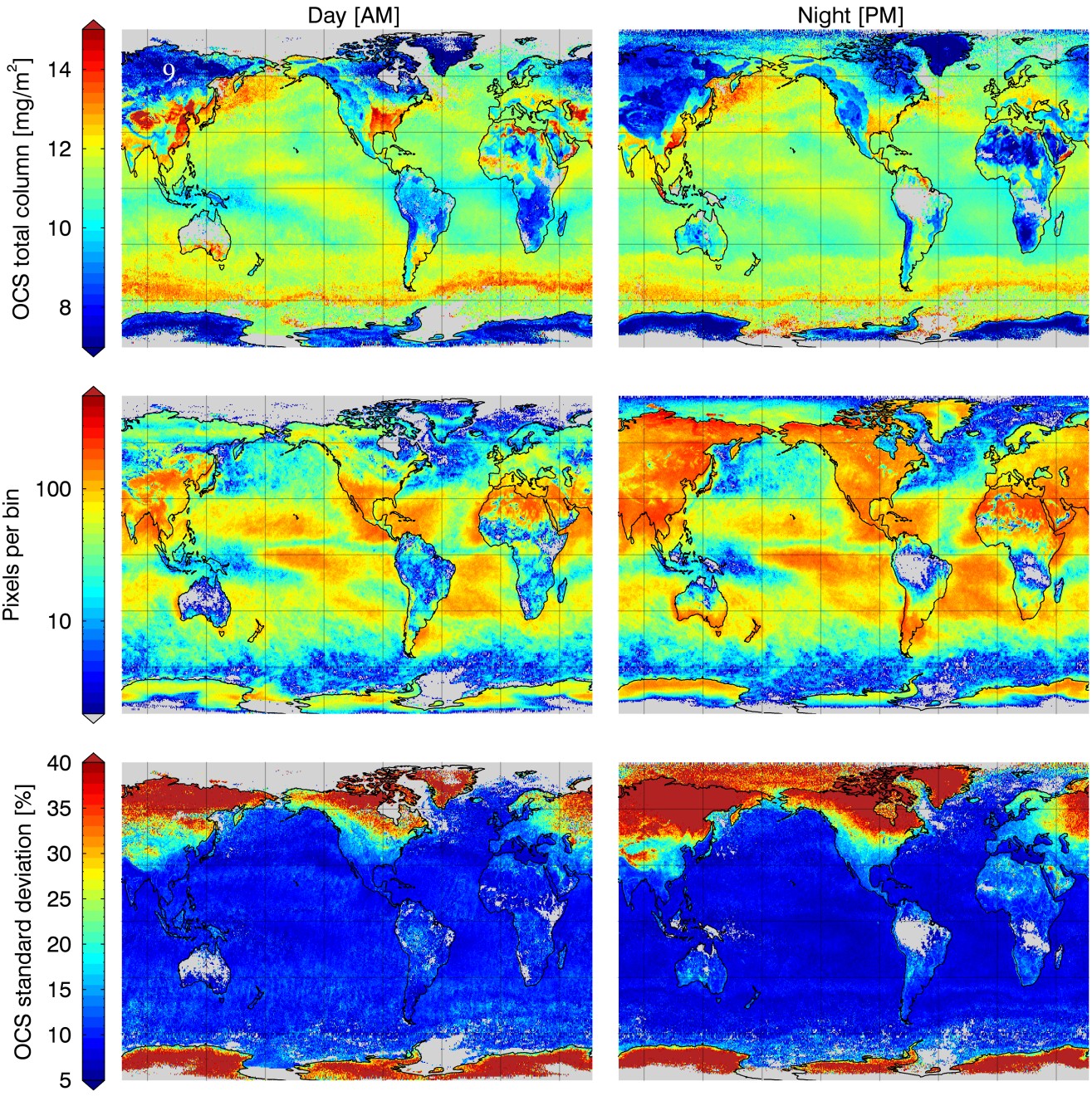

**Figure 10.** January–February 2014: Linear estimates of total column OCS median values are shown in the top row for sun illuminated morning (left column) and night-time evening (right column). The results are binned by latitude-longitude widths of $0.5° \times 0.5°$. The middle row shows the number of pixels per spatial bin that passed the quality control checks. The bottom row shows the sample standard deviation of OCS per bin for the two month interval. Spatial bins with missing data are coloured grey.

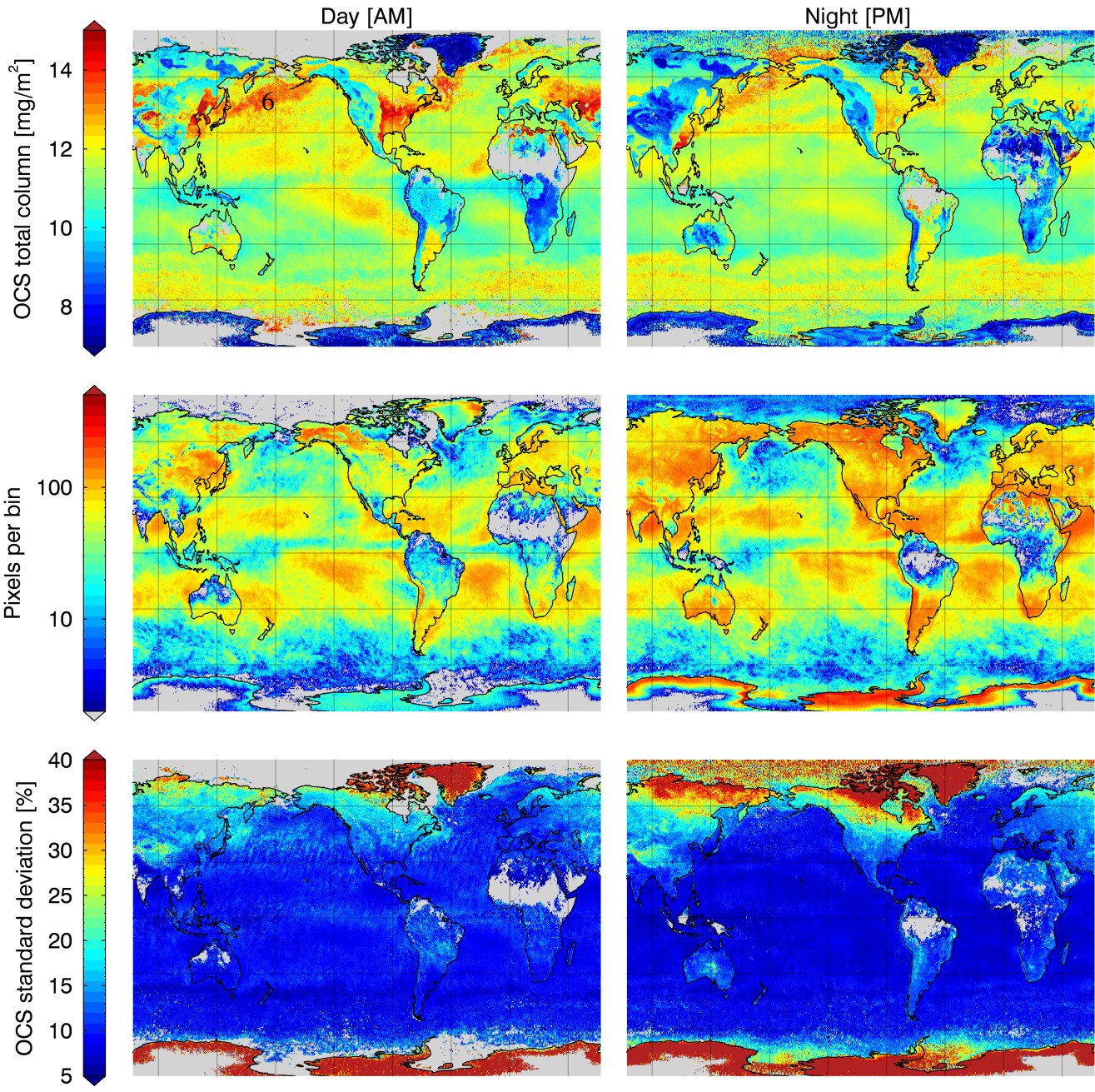

**Figure 11.** Same as in Fig. 10, but for March–April 2014.

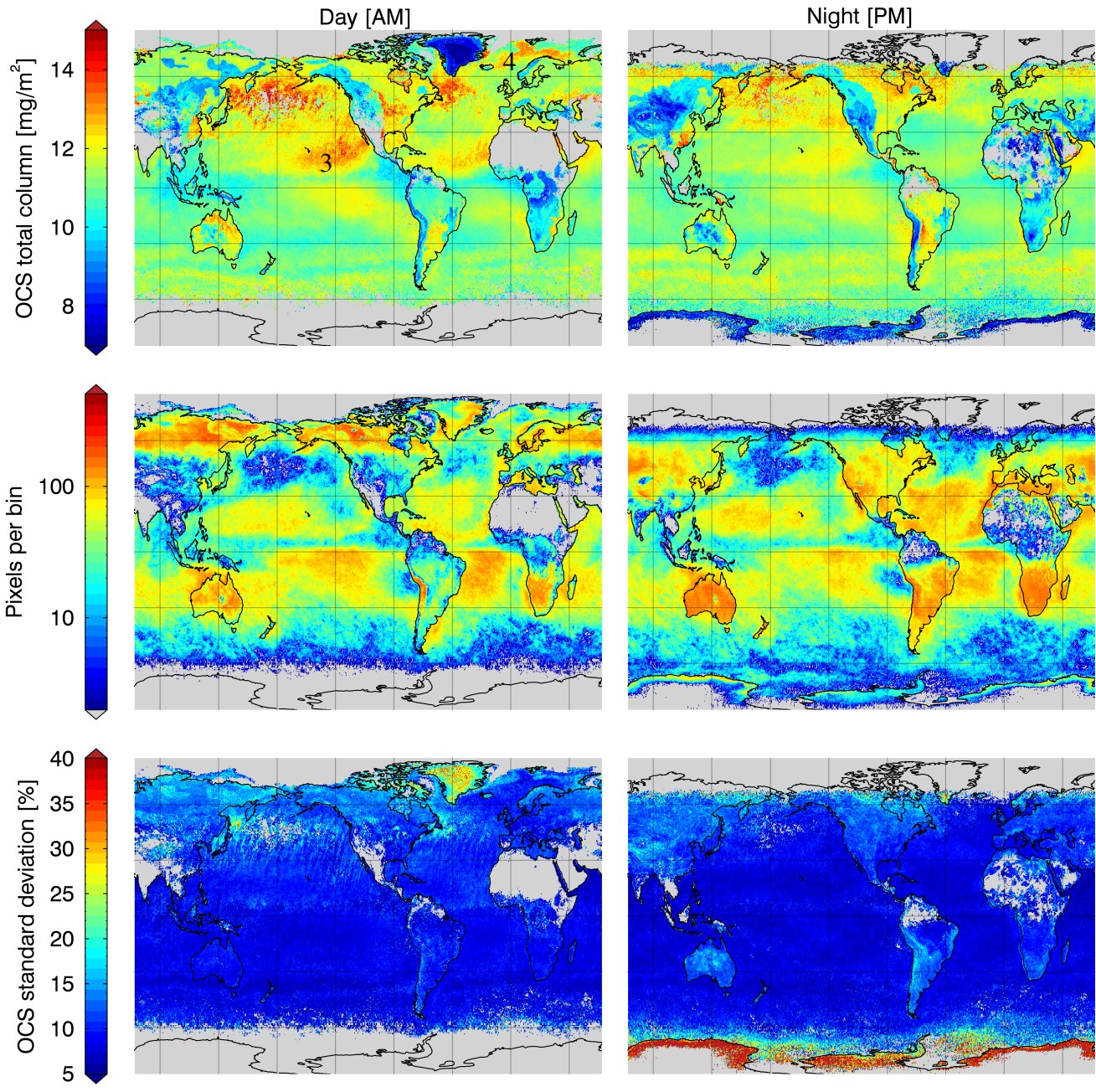

**Figure 12.** Same as in Fig. 10, but for May–June 2014.

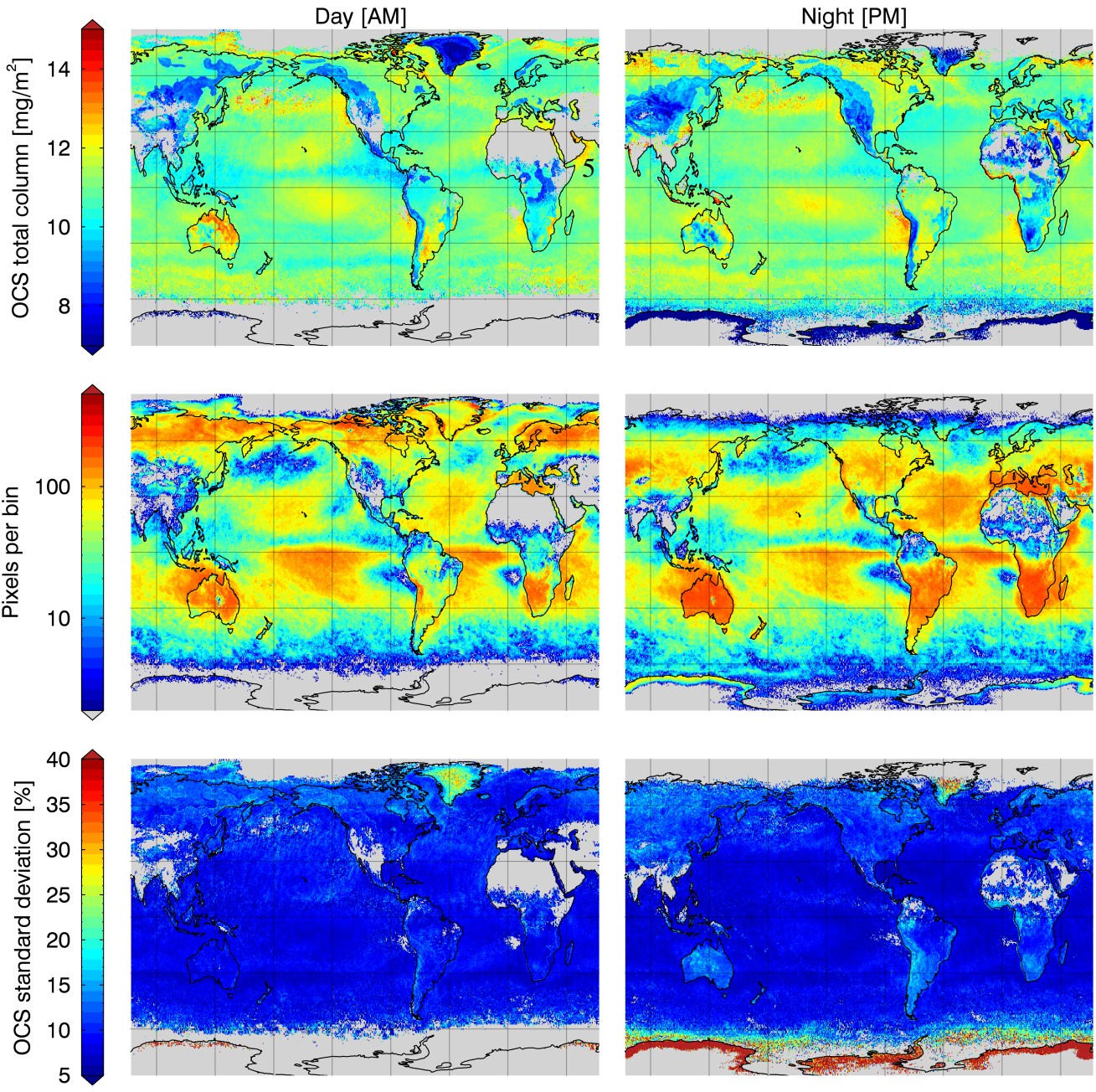

**Figure 13.** Same as in Fig. 10, but for July–August 2014.

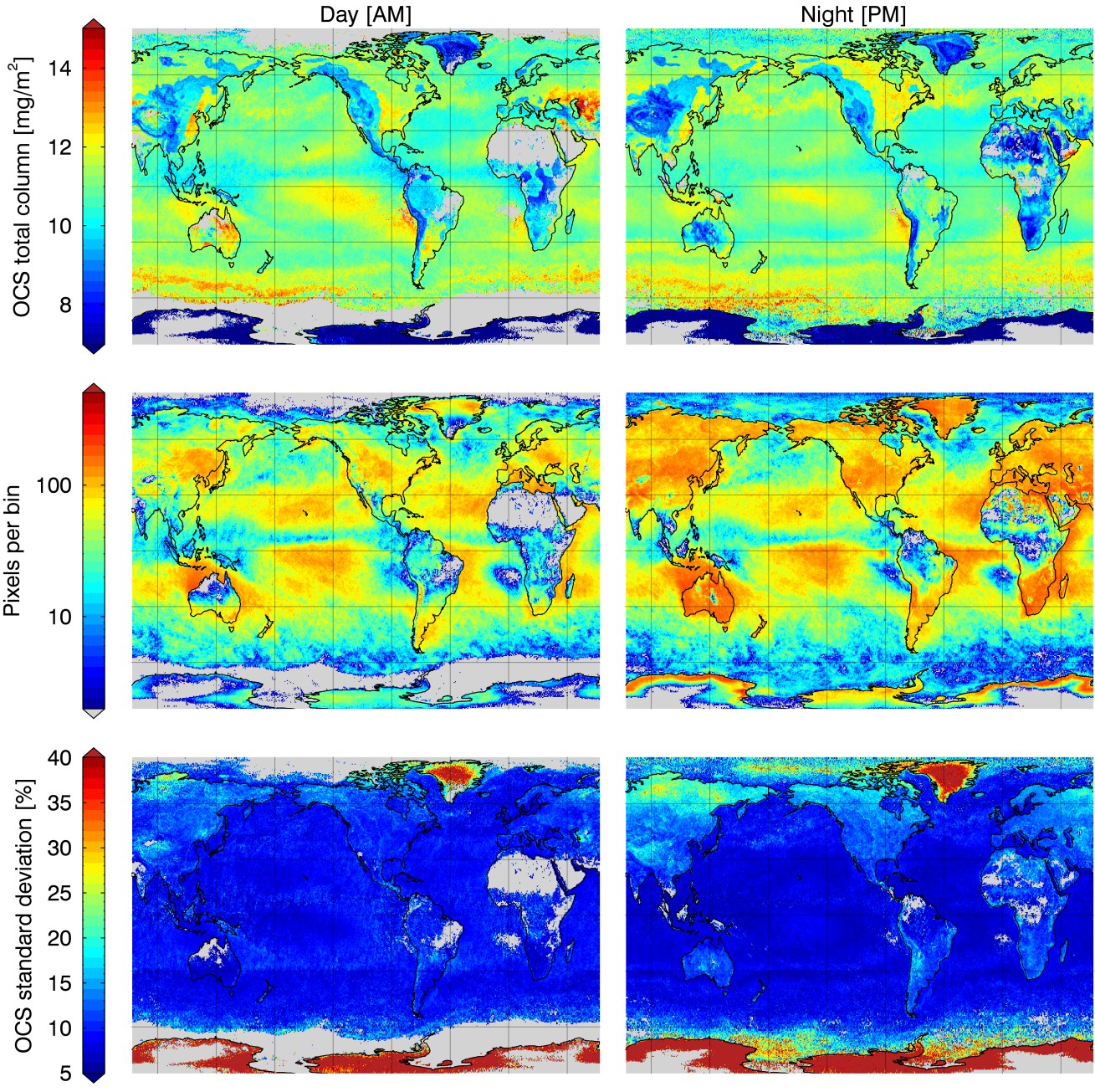

**Figure 14.** Same as in Fig. 10, but for September–October 2014.

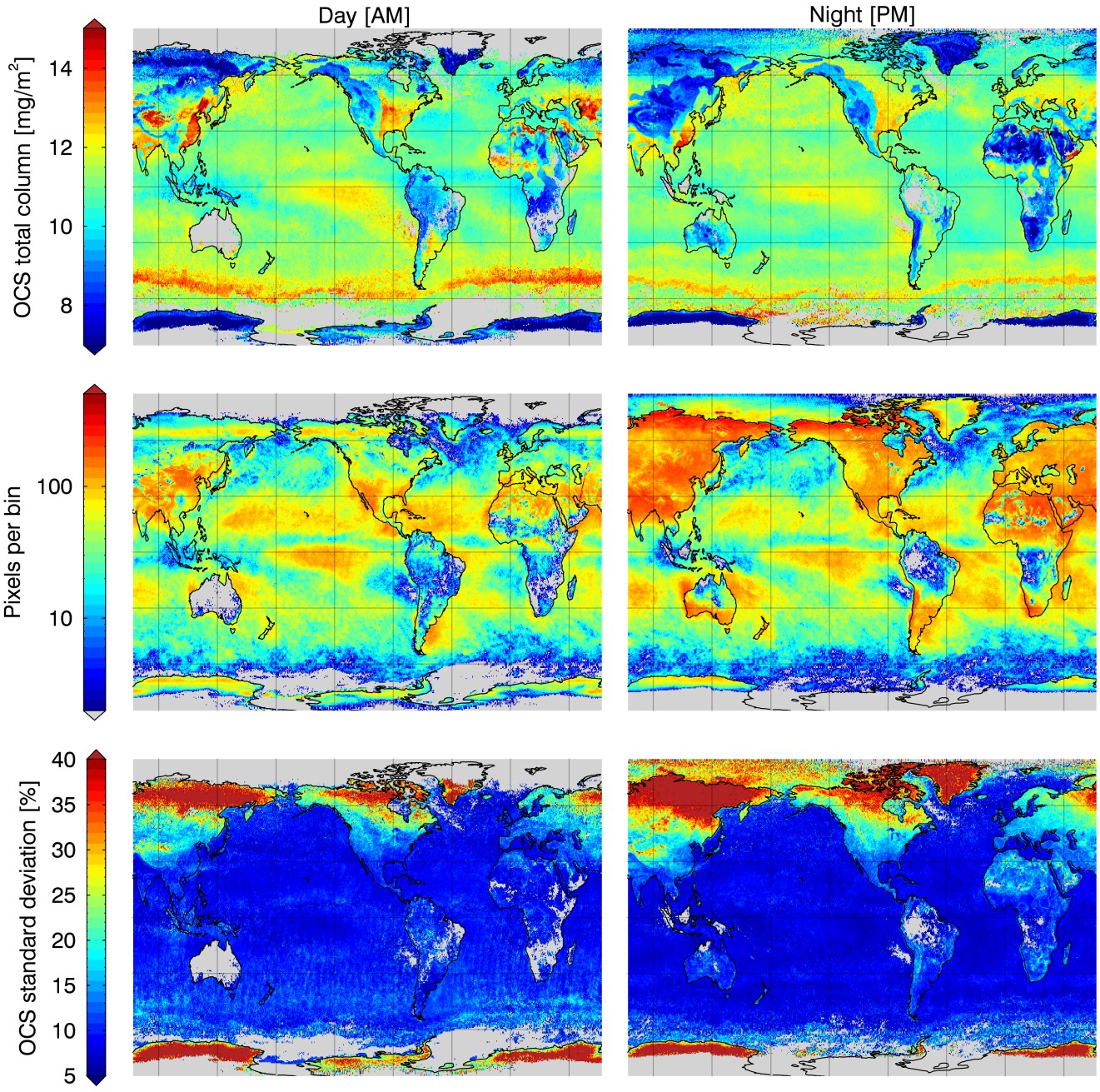

**Figure 15.** Same as in Fig. 10, but for November–December 2014.

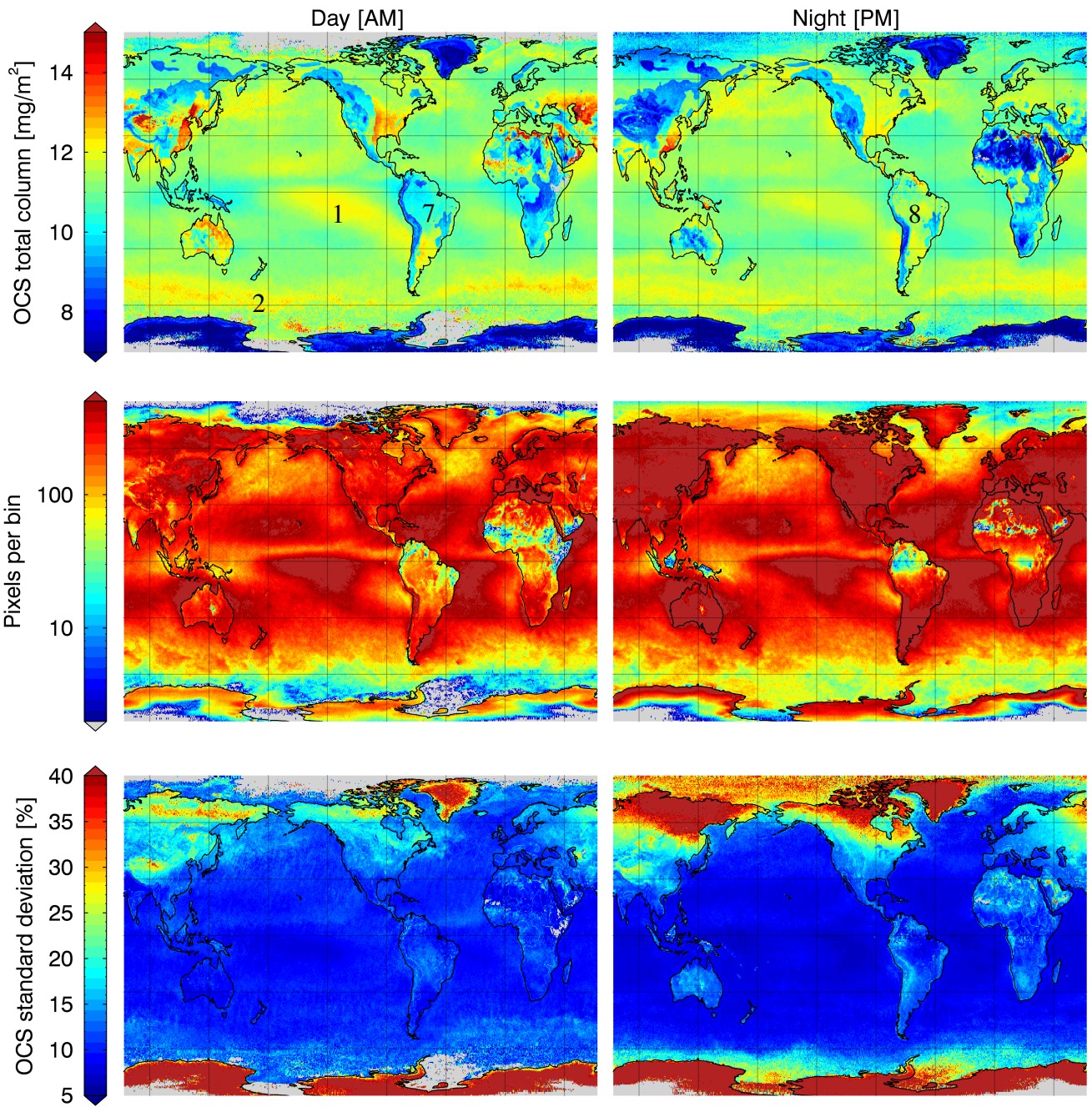

**Figure 16.** Same as in Fig. 10, but for all of 2014 combined.

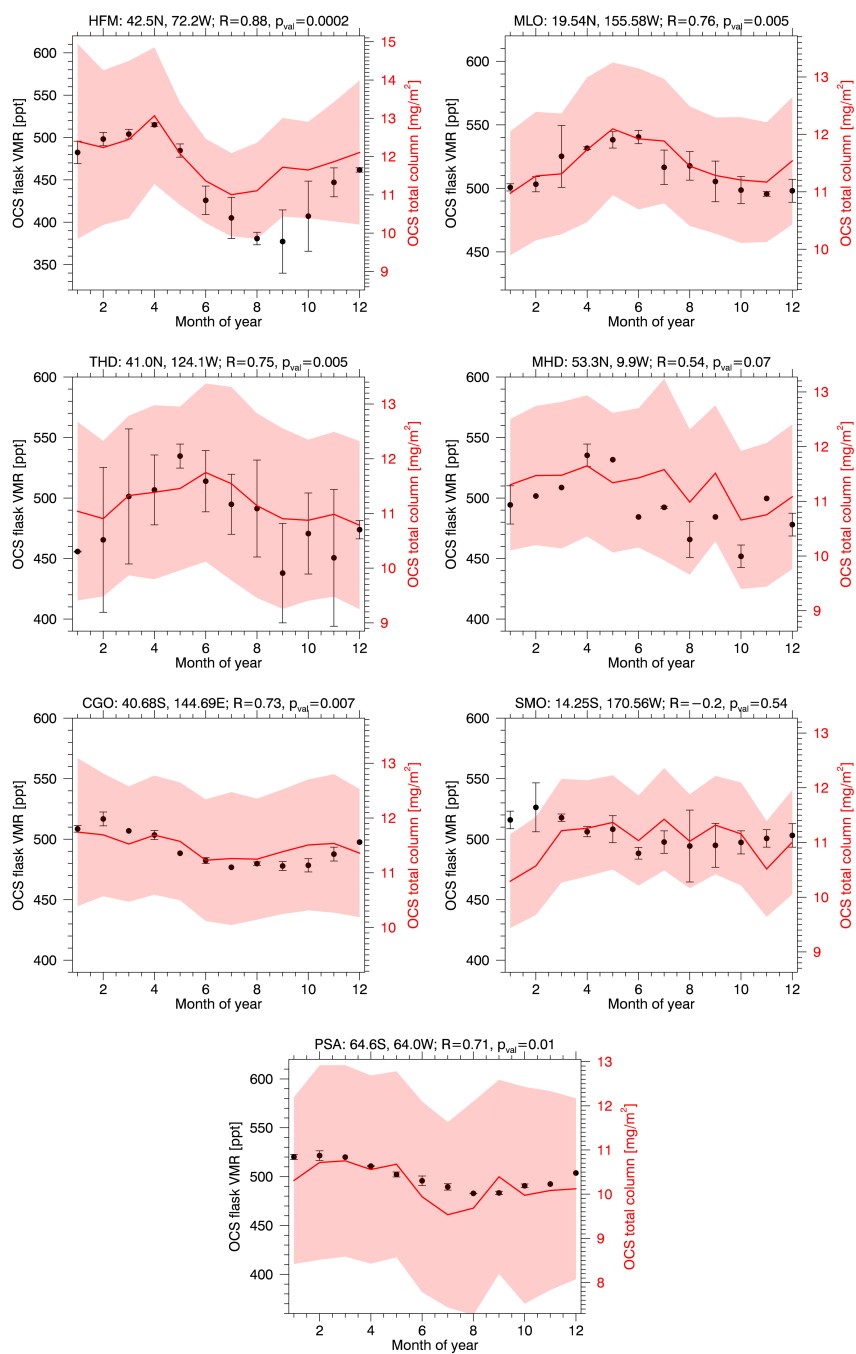

**Figure 17.** OCS total column median estimates (red) from the linear retrieval are compared to NOAA flask measurements of OCS surface VMR (black) binned by 12 month increments throughout 2014. Retrieval estimates are taken from a $2°$ radius about the NOAA site locations. The shaded red area represents the sample standard deviation of the total column estimates and the black error bars are the standard deviation of the flask samples within that month (*not* divided by the square root of the sample size).