# Peer review of "Fast retrievals of tropospheric carbonyl sulphide with IASI"

_Atmospheric Chemistry and Physics, 2016_

## Referee Comment (RC1) · Anonymous Referee #1 · 7 Oct 2016

This paper, presents a new way of doing linear retrievals from infrared satellite observations. The method is applied on OCS retrievals from the IASI instrument, but is equally adaptable to the retrieval of other species from other instruments. The paper ends with the presentation and discussion of the generated dataset of retrievals of one-year of OCS measurements.

In many ways the paper is exemplary: it is well-written, clear and innovative. This is especially true for the first part of the paper where the retrieval technique is detailed -it makes for very interesting reading. There is obviously room for improvement in some areas of the algorithm (for instance in the treatment of surface properties). The richness in ideas more than compensates this lack of maturity. The second part, the discussion of the retrieval results of the 2014 IASI data is less deep, but demonstrates that the algorithm is capable of capturing at least some of the global/seasonal OCS

variability. Although given the maturity of the product, caution should be exercises in drawing conclusions from the data as several identified enhancements/depletions are clearly related to problems in the retrieval. This is acknowledged by the authors but could be emphasized more clearly in places. Presentation-wise, the second part is also less strong than the first, and below, several improvements are suggested. Overall, I deem this paper to be suitable for publication in ACP after a minor revision.

Comments are listed in chronological order:

P1, L6. I would remove "rather than treated as effective noise". This is already pretty technical and few readers probably would understand this before having read the rest of the paper.

P1, L24. "fast retrieval methods". The use of a computer cluster is also a viable alternative.

P2, L3. a comma is missing after dramatically (sentence might need to be reworded, the 'indeed' sounds colloquial)

P4, L13/14. These are definitely not the main reasons. TES does not cross scan like IASI (it has no swath) and also has huge gaps in between two nadir pixels. I do not have the numbers at hand but the number of TES observations is several orders smaller than IASI's 1 million+ spectra per day.

P4, L21. "reduce" should probably be "reducing".

P7, F1. What does BBT stand for?

P7, F1. This figure would in my opinion be more useful if it showed the actual contribution of each of the species in the IASI spectrum rather than the jacobians. That is: for the bottom plots to plot the difference between the simulated spectrum and the simulated spectrum without the different individual species included in the forward model. That way, the individual contribution of each of the species is highlighted very clearly, and the reader can see the extend to which OCS, O3, etc. contribute to the

IASI spectrum.

P8. Clearly, and that I think is nowhere mentioned in the manuscript, it is actually not necessary to calculate G * y for all rows of G. If one is just interested in OCS, it would suffice to carry out the multiplication of the first row only. But of course, the first row in the matrix G is affected by the jacobians of the other parameters. I think it could benefit the reader to discuss this in short.

P8. How much does it really help to retrieve all these parameters along with OCS? Have you tried just retrieving OCS and carrying out the channel selection based for this? The retrieval of this full state vector is one of the main innovative aspects and it would therefore be well worth exploring/explaining/illustrating this further. It would in any case be more convincing if the results could be compared with or without retrieving a full state vector.

P11, L3. Levenberg-Marquardt method, please add a reference to the exact method which was used here (as the specific iterative procedure is not discussed)

P11, L23. "In reality", this confuses me as I feel I am missing something. It thought this followed naturally from the above? Please expand.

P13, L9-10. Is the same not done for the $CO_2$ q branch (P14, L23). One could do this for the each channel thereby reducing the sensitivity to errors in the forward model.

P14. On which atmosphere was this analysis carried out?

P14. L34. "mean spectrum" what is the meaning of "mean" here. From what was said before 80 atmosphere yield 80 spectra, so I am not sure what is being averaged here - the term mean spectra is also used in several places afterwards.

P15 L5-10. This could be more clear. First, "x0" and "y0", shouldn't these be xj and yj, etc...with j=1..80? Then Chi_pr, G and K should all have an index j too, since they also depend on the specific atmosphere. Secondly, it would be good to show the extra step here (ie. eq (10) with Eq (2) substituted), I found this section especially confusing on a

first read, and that extra step would have helped.

P15 L9. "In some texts". Please give an example, what is the DRM generally used for or what does it in general represent?

P16. A fourth obvious approach is not listed here, that is to select the atmosphere based on closeness in time and location of the observed spectra and the time/location of the 80 reference atmospheres. I think it is worth to discuss this approach in short.

P17. F9. There seems to be bias in the linear assumption error. Do you have an idea why this is so? I see no reason why this would be so (the a priori is unbiased), so they should all four have been nicely spread around 0, but with a difference in spread?

P17 L4. shouldn't "three selection methods discussed" read "the three discussed selection methods"? (as a non-native English speaker I am unsure)

P18. L1/2. Why is that so? Did you try with even less channels? Clearly reducing the number of channels improves the chance of relying on badly modeled channels, but the channel selection procedure assumed a perfect model; so I see no obvious reason why this would be. It is a very interesting finding, but would be good if you could expand on the underlying reasons. This comes back in the conclusion (twice) and is each time stated, but the underlying reason is never given.

P18. L21/25 Five thermal contrast scenario's seem little. Thermal contrast can go up to 30 K in favorable circumstances. This is one of the places, where the retrieval algorithm could easily be improved.

P18. L29. But from what follows it seems chi_projected is calculated (equation 14). The two should be identical no?

P19. L6. Can you give (or at least cite) the exact formula which was used to calculate the specular solar reflection angle?

P19. L19. "Thus" the factor 2 doesn't strictly follow from what is written above. Perhaps

better would be "Thus, a reasonable criteria for accepting a ..."

P19. Section 4. There are three ways which I think would improve the presentation drastically. Firstly, 36 figures 6 x Fig10-15 are too much, especially since most of these bimonthly maps are not discussed. I would strongly suggest replacing these 36 figures, with a one page 4 x 2 panel figure showing just the OCS panels for AM/PM for the 4 seasons. This would allow to compare much easier the different seasons. All the other panels aren't that useful, and most of what is seen in them can also be seen on Figure 16, which can be kept in its current form.

P19. Section 4. The second presentation suggestion I have, is to display OCS as a VMR on a fixed altitude (altitude of max sensitivity?). Currently it is very difficult to interpret the columns over land because of orography. The maps now basically look like earth surface ground height maps. Satellites are of course sensitive to a column rather than vmr at a given location, still as only one parameter is retrieved and the whole profile is scaled uniformly, it really doesn't harm to show the value at one given altitude (even though care much be taken not to over-interpret those values of course). Orographic effects should be far less visible that way and in addition it would also be much clearer whether the retrieval sees an enhancement or depletion with respect to the a priori (the apriori could be indicated on the colorbar).

P19. Section 4. Thirdly, it would be nice to show a modeled plot of OCS vmrs to represent 'the state of the art' of the current knowledge on OCS distributions. This would greatly ease discussion (it could first be discussed in section 2, and then referred to in section 4).

P20. L4. "likely due". This would be very easy to check no? In fact, it wouldn't be too hard, and quite instructive to produce a map which for each place on Earth shows the filter which was most often applied.

P20. Section 4.1. One thing that occurred to me was that the daytime ocean seems to have higher highs and lower lows, can you confirm/explain?

P21 L4. Please add a number/point, as was done for the other points of interest.

P21/22. I would personally be even more cautious in over-interpreting the data, given the maturity of the product.

P23. L14. "import" should probably be "important"

---

## Short Comment (SC1) · 7 Oct 2016

It is very exiting to see IASI OCS data. I have a few comments after I look at the figures.

1) The OCS total column maps always show a wired feature that a sharp change from west US to east US with a smooth line coming down from north to south. This is quite suspicious and may related to scan angle.

2) Also IASI OCS in this paper is showing total column, IASI should be most sensitive to mid tropospheric OCS, why authers do not show how it is compared with HIPPO OCS across Pacific? Validation with NOAA ground measurements is great but not enough.

Hope my comments would help to improve the study.

---

## Short Comment (SC2) · 10 Oct 2016

1. Thanks author to help me better understand the feature over US. Yes, showing IASI OCS in VMR will be better for the data to be used to understand the OCS global budget. I wonder if IASI would retrieve column O2 then it would be helpful to get column VMR and reduce the topographic effect.

2. To compare to HIPPO data, IASI data do not have to be in the same year but the same month of the year would be fine if consider the interannual variability is smaller than latitudinal gradient.

For the FTIR measurement comparison, please keep in mind, NDAC and TCCON OCS would be available in near future. Hope these data would provide more cross check.
At the end, I'm very glad to hear IASI could provide OCS observations and provide a new source of satellite measure in future. I'm also look forward to OCS from follow-on IASI instruments to provide a longer record.

---

## Author Comment (AC1) · 10 Oct 2016

Dr. Kuai, thank you for your interest and comments on this paper. Here is a brief response as further discussion will be rolled into one response compiled from both referees.

1. The wired feature you mention is not due to scan angle, but rather relates to the geographical altitude (orography) of the region being observed. The figures shown in this paper are of OCS total column amount, so the vertical profile of OCS has been integrated with respect to pressure. As mentioned by the first referee, another option would have been to display these estimates as a representative OCS volume mixing ratio (VMR) at a certain level, e.g., $500$ hPa. However, there is simply not enough information in the observation to resolve vertical structure of OCS using IASI. While displaying a representative VMR removes orography, it may mislead readers unfamiliar with the nuances of information theory applied to atmospheric retrievals and have them believe the vertical resolution is much greater than it really is. Therefore, I prefer to display total column, because this is what is actually being estimated and it forces the reader to contemplate this point.

2. Yes, a comparison to the HIPPO campaign would be great. However, the method developed in this paper uses the AVHRR cloud fraction product to discriminate between clear and cloudy scenes. Unfortunately, the AVHRR cloud fraction was embedded in the IASI Level-1c data towards the end of 2011 and just missed the last HIPPO flight. Introducing a second data stream from AVHRR separately requires a significant rewrite of the retrieval algorithm, which may be attempted in future work.

   Additionally, I tried to find other estimates of OCS total column taken from ground-based FTIR spectrometers. However, accessing this data required personal permission from approximately one dozen different scientists. Therefore, the ground samples taken by Steve Montzka and the NOAA team were the only *in situ* data accessible to most of the scientific community from 2014.

   Finally, I will point out that the NOAA site at Mauna Loa samples the atmosphere at an altitude higher than 13,000 feet; close to peak IASI sensitivity as you mentioned. Notice that the seasonal correlation between the IASI linear estimates and *in situ* samples at Mauna Loa have a correlation of 0.76.

Thank you again for your comments.

---

## Short Comment (SC3) · 12 Oct 2016

I welcome this paper since it bravely deviates from the usual and common implementation of Optimal Estimation (OE). In doing so, the paper largely agrees with retrieval strategies so long pursued within IASI activities by my group and colleagues of mine.

Having said that, I have to stress that the fact that OCS can be retrieved with a poor a prior constraint and based only on IASI data has been recently put forward by Liuzzi et al 2016 (doi:10.1016/j.jqsrt.2016.05.022), who demonstrates and exemplifies how OCS column amount can be retrieved with a degree of freedom of 1 and a precision for the single IASI field of view of 6-7%. In this respect, the paper by Vincent and Dudhia is lacking important references and should be updated.

The "unprecedented features" (pag, 2 lines 7-17) claimed in the paper are common practice within the IASI community.

The use of *ensemble of states* to initialize the non-linear retrieval scheme dates back to Chedin et al, *J. Climate Appl. Meteor.,24, 128-143, 1985* (TIGR, Thermodynamic Initial Guess Retrieval, http://ara.abct.lmd.polytechnique.fr/index.php?page=tigr). Furthermore, the approach has been used in the context of OE to linearize the radiative transfer equation by (to name a few) Grieco et al 2007 (doi:10.1002/qj.162), Masiello et al 2009 (doi:10.5194/acp-9-8771-2009 ) who used an ensemble of states to train an EOF regression for the purpose of computing a suitable background around which to linearize the inverse problem, instead of simply using the *ensemble-mean atmosphere*. This approach, i.e. EOF initialization, would greatly improve the algorithm shown in Vincent and Dudhia paper, e.g., to improve the linearity in $H_2O$ channels. Presently, Liuzzi et al 2016 linearizes the OE retrieval with time-space collocated ECMWF analysis.

The use of a *data driven retrieval approach* which simultaneous (*jointly*) retrieves temperature and gases dates back to Smith et al, 1991 (http://dx.doi.org/10.1364/AO.30.001117) and has been extended by Liuzzi et al 2016 to simultaneously (jointly) retrieve emissivity, surface temperature, $H_2O$, HDO, $O_3$, $CO_2$, $N_2O$, CO, $CH_4$, $CF_4$, $SO_2$, $NH_3$, $HNO_3$, and finally OCS. Previous specific application to gas retrieval alone can be found in Grieco et al 2013 (doi:10.1364/OE.21.024753).

As I said, this paper provides an interesting approach to the design and implementation of a non-conventional OE data driven retrieval strategy, which is specifically designed with a loose a-prior constraint. In this respect, authors should be encouraged. However, they should correctly acknowledge work already done in the same area. In addition, there are many weak points that should be properly addressed.

The authors fails to show how the retrieval products are correlated, one would rather expect a strong correlation between OCS and $T_s$ since the authors seem to privilege atmospheric window channels. Beyond 2000 $cm^{-1}$; radiances (especially window channels) are contaminated by solar radiation. Authors avoid sunglint, however solar contamination is in any case to be taken into account in daytime. Over land, especially for desert, arid and semi-arid regions, the emissivity is strongly dependent on the wavenumber, so that the use of an effective $\varepsilon T_s$ is not adequate. The $CO_2$ spike at 2077 $cm^{-1}$ (Fig. 5 in the paper) is not seen in state-of-art forward modelling calculations (e.g., once again see Liuzzi et al 2016). Therefore, how accurate is authors' RFM? It is not clear how the radiative transfer is manipulated to arrive at the state vector defined in Eq. (7). The state parameters seem to have been included rather ad hoc, whereas the state vector should be derived by a coherent linearization of the radiative transfer equation, accounting for the physical parameters that contribute to the radiance signal and showing how these can be transformed to the state vector of Eq. (7). Finally, what about CO and $O_3$ retrievals? Why they are not shown.

---

## Author Comment (AC2) · 23 Oct 2016

Thank you for the constructive comments. We will review the recommendations and upload the responses as a supplement to the article upon resubmission.

---

## Referee Comment (RC2) · Anonymous Referee #2 · 15 Nov 2016

**1 General Comments**

This paper presents a year of global space-based measurements of OCS using a sophisticated linear retrieval method. The paper is very well written and clear and the conclusions the authors draw are reasonable. This paper represents a valuable addition to the capability of satellite based sensors to resolve OCS. The method in this paper furthermore allows for faster retrievals than have been possible previously allowing for rapid analysis over long periods. The measurements of OCS they have managed to derive are impressive. They have put a lot of effort into characterising potential sources of systematic error related to the retrieval method and include a balanced discussion of the strengths and weaknesses of the approach and the expected impact on the results.

**2 Specific Comments**

pg 2 l 5 Consider clarifying to indicate that the random error associated with the mean or median value can be improved by averaging.

pg 3 l 27 where you mention "manned space flight" would be good to mention the mission name.

pg 5 l 14 would say here that $\epsilon$ is the error in measured signal relative to the linearised forward model rather than just forward model.

fig 2 pg 8 are these two plots the different surface temperature contrast scenarios? Needs mentioning in the figure caption.

pg 12 l 5 Perhaps you mean "will show an associated spectral feature beyond the standard deviation" rather than "in the standard deviation" here?

pg 13 l7 Perhaps the part about how you deal with spectral variations due to H2O in the measurement error covariance could be illustrated by an equation? For the nomenclature, here do you really mean the "variances of the measurement covariance" or do you mean both the covariance and variance entries of that matrix. Where it states "measurement covariance", this strictly refers to $< yy^T >$ i.e., the covariance of the measurement rather than the "measurement error covariance" $< \epsilon\epsilon^T >$ although you don't always see them distinguished in the literature. Here do you strictly mean measurement error covariance?

pg 16 l 5 Here it's that are only 80 independent atmospheres that implies there should be colinearilty in the overdetermined system in Eq 12 which means you expect to be able to reduce the dimensionality. Therefore perhaps reconsider the wording of the sentence "However, since there are only 80 independent atmospheres considered, the dimensionality of the problem must be reduced....".

pg 17 Found it somewhat difficult to follow which methods were being referred to in

the discussion about the method to select the initial atmosphere. Perhaps you could specifically label the methods to help the reader follow more easily.

---

## Author Response (AR1)

**Final Author Response**

**December, 2016**

Here are the responses to the first and second referee's comments as well as those provided by Italia De Feis. Responses to Le Kuai's comments were uploaded into the discussion forum.

We thank all the commenters for their interest and feedback in this paper.

1. **P1.L6. I would remove "rather than treated as effective noise". This is already pretty technical and few readers probably would understand this before having read the rest of the paper.**
   Whether the reader fully understands this point in the abstract or not, it sets an upfront contrast between how this methodology is implemented compared to other recent papers. There are really only two viable options for dealing with systematic errors without ignoring their effects, which are to model them jointly or treat them as effective noise. While most readers will not appreciate this point when reading the abstract, they will at least be aware there is an alternative option and read further into the paper for more clarity.

2. **P1.L24 "fast retrieval methods". The use of a computer cluster is also a viable alternative.**
   This is true if you have the money and resources available. I added a brief comment about this possibility (see tracked changes).

3. **P2.L3 A comma is missing after dramatically (sentence might need to be reworded, the 'indeed' sounds colloquial)**
   Agreed. The sentence was reworded.

4. **P4.L13/14 These are definitely not the main reasons. TES does not cross scan like IASI (it has no swath) and also has huge gaps in between two nadir pixels. I do not have the numbers at hand but the number of TES observations is several orders smaller than IASI's 1 million+ spectra per day.**
   Thanks for pointing this out. The sentence was modified to highlight the fact that cross-scanning provides greater spatial sampling for IASI.

5. **P4.L21 "reduce" should probably be "reducing".**
   I believe the verb reduce is appropriate here.

6. **P7.F1 what does "BBT" stand for?**

BBT stands for blackbody brightness temperature and is now defined in the first paragraph of Sec. 3.2.

7. **P7.F1 This figure would in my opinion be more useful if it showed the actual contribution of each of the species in the IASI spectrum rather than the jacobians. That is: for the bottom plots to plot the difference between the simulated spectrum and the simulated spectrum without the different individual species included in the forward model. That way, the individual contribution of each of the species is highlighted very clearly, and the reader can see the extent to which OCS, O3, etc. contribute to the IASI spectrum.**

This is technically also a "Jacobian," (delta signal/delta VMR) where the change in VMR goes from 0 to a standard atmospheric amount. The figure as it is shows a change in signal (BBT) for a 1% increase in VMR. Aside from the magnitude of the y-axis, the spectral shapes for each gas will change slightly, but I doubt any additional insight will be gained by modifying the figure this way.

8. **P8. Clearly, and that I think is nowhere mentioned in the manuscript, it is actually not necessary to calculate G * y for all rows of G. If one is just interested in OCS, it would suffice to carry out the multiplication of the first row only. But of course, the first row in the matrix G is affected by the jacobians of the other parameters. I think it could benefit the reader to discuss this in short.**

This is a good point. The first paragraph of Sect. 3.3 was modified to address this comment. The reason for carrying through with the joint estimate is that the other parameters are used for retrieval diagnostic information later on for down selection of low confidence pixels.

9. **How much does it really help to retrieve all these parameters along with OCS? Have you tried just retrieving OCS and carrying out the channel selection based for this? The retrieval of this full state vector is one of the main innovative aspects and it would therefore be well worth exploring/explaining/illustrating this further. It would in any case be more convincing if the results could be compared with or without retrieving a full state vector.**

I agree that it would be useful to compare this technique along with numerous other linear methods to analyze the strengths and weaknesses of each. However, the space this opens for exploration is huge. We could take three techniques that span the retrieval method space and focus on those. However, this would greatly increase the length of the paper and shift the focus to technique comparisons with an epilogue of OCS results. I think this question should be addressed more robustly in a follow-on paper that would be more appropriate for publication in AMT or IEEE Geoscience and Remote Sensing.

10. **P11, L3. Levenberg-Marquardt method, please add a reference to the exact method which was used here (as the specific iterative procedure is not discussed)**

    The method is cited in the text. The description provided by Rodgers in Ch. 5.7 is the method used here. Everything the reader may want to know about the Levenberg-Marquardt method used for this section is described by the referenced document.

11. **P11, L23. "In reality", this confuses me as I feel I am missing something. It thought this followed naturally from the above? Please expand.**

    The ending of this paragraph was modified to explain that the criteria for setting the state vector and prior covariance introduced in the previous section was convergence of the test-bed iterative model along with the spectral residual between model and measurement.

12. **P13, L9-10  Is the same not done for the CO2 q branch (P14, L23). One could do this for the each channel thereby reducing the sensitivity to errors in the forward model.**

    All diagonal elements in the prior covariances are scaled as mentioned, not just the H2O channels. The last sentence of the paragraph was updated to reflect this.

13. **P14. On which atmosphere was this analysis carried out?**

    The captions to Figure 7 and 8 state that it is a mid-latitude atmosphere.

14. **P14. L34. "mean spectrum" what is the meaning of "mean" here. From what was said before 80 atmosphere yield 80 spectra, so I am not sure what is being averaged here - the term mean spectra is also used in several places afterwards.**

    The previous sentence really defines what is meant here and this sentence is just a simplified description. However, "(i.e., averaged along the spectral axis)" was added to the text.

15. **P15 L5-10. This could be more clear. First, "x0" and "y0", shouldn't these be xj and yj, etc...with j=1..80? Then Chi_pr, G and K should all have an index j too, since they also depend on the specific atmosphere. Secondly, it would be good to show the extra step here (ie. eq (10) with Eq (2) substituted), I found this section especially confusing on a first read, and that extra step would have helped.**

    Yes, the presentation of this was a bit sloppy. An index j was used instead of 0 and a middle step was included as recommended.

16. **P15 L9. "In some texts". Please give an example, what is the DRM generally used for or what does it in general represent?**

    Rather than discuss what other texts refer to as the DRM, mention of this was removed and KG is simply presented as is. While GK has clear physical meaning in constrained retrievals, KG is a more abstract concept best left as a mathematical construct resulting from the development of a projected cost.

17. **P16. A fourth obvious approach is not listed here, that is to select the atmosphere based on closeness in time and location of the observed spectra and the time/location of the 80 reference atmospheres. I think it is worth to discuss this approach in short.**

Yes, this is another viable selection method for choosing an initial atmosphere. However, the RTTOV ensemble is not appropriate for this method due to its irregular sampling. A paragraph was added to address this point.

18. **P17. F9. There seems to be bias in the linear assumption error. Do you have an idea why this is so? I see no reason why this would be so (the a priori is unbiased), so they should all four have been nicely spread around 0, but with a difference in spread?**

For the two that matter in this paper, there is no noticeable bias in the linear assumption error. The third simply did not work well and the distribution from random selection (the baseline) would require far more than 80 atmospheres to observe a normal distribution with the observed spread. Perhaps discussing this in detail is not particularly useful to the reader and increases length with little gain.

19. **P17 L4. shouldn't "three selection methods discussed" read "the three discussed selection methods"? (as a non-native English speaker I am unsure)**

Corrected as recommended.

20. **P18. L1/2. Why is that so? Did you try with even less channels? Clearly reducing the number of channels improves the chance of relying on badly modeled channels, but the channel selection procedure assumed a perfect model; so I see no obvious reason why this would be. It is a very interesting finding, but would be good if you could expand on the underlying reasons. This comes back in the conclusion (twice) and is each time stated, but the underlying reason is never given.**

A few sentences were added to this paragraph that state that channel selection is an important step in reducing the effect of systematic errors, such as neglecting non-linearity. If the retrieval were ideal, then adding more channels would always increase information. However, adding channels of minor importance only increase sensitivity to systematic errors in the imperfect retrieval (as all are to some degree).

21. **P18. L21/25 Five thermal contrast scenario's seem little. Thermal contrast can go up to 30 K in favorable circumstances. This is one of the places, where the retrieval algorithm could easily be improved.**

Agreed, but it is not quite that simple. In this method we expanded the searchable ensemble to include these thermal contrast scenarios over land. It would be better to estimate thermal contrast from the signal and use that to reduce the larger ensemble to an

appropriate subset. I have some ideas of how to do that, but for now this improvement must be left for future work.

22. **P18. L29. But from what follows it seems chi_projected is calculated (equation 14). The two should be identical no?**

    No, the result from an iterative retrieval will only be the same as the first step if the problem is linear. The projected cost mentioned will likely be greater than the converged cost if the problem is non-linear.

23. **P19. L6. Can you give (or at least cite) the exact formula which was used to calculate the specular solar reflection angle?**

    Cited a PhD thesis where this calculation is discussed as a section within a chapter.

24. **P19. L19. "Thus" the factor 2 doesn't strictly follow from what is written above. Perhaps better would be "Thus, a reasonable criteria for accepting a ..."**

    Agreed, the sentence was modified in accordance with the recommendation.

25. **P19. Section 4. There are three ways which I think would improve the presentation drastically. Firstly, 36 figures 6 x Fig10-15 are too much, especially since most of these bimonthly maps are not discussed. I would strongly suggest replacing these 36 figures, with a one page 4 x 2 panel figure showing just the OCS panels for AM/PM for the 4 seasons. This would allow to compare much easier the different seasons. All the other panels aren't that useful, and most of what is seen in them can also be seen on Figure 16, which can be kept in its current form.**

    In general, I agree with this comment about streamlining the presentation of the results if the paper were submitted to a page-limited journal, such as Science, GRL, or even JGR. The results were presented this way because there is space available in an electronic only format to be generous with the amount of information portrayed to the reader. Bi-monthly maps are presented instead of seasonal maps to highlight the important fact that the retrieval SNR is high enough to resolve interesting structure at temporal scales finer than what is normally presented. Additionally, I would insist that even if going to a seasonal representation that the number of observations and sample standard deviations be included as well, because any estimate without some form of displayed uncertainty is meaningless. The yearly sample size and sample standard deviations do not accurately reflect the bi-monthly variations. There is a trade off between portrayed information and succinct readability. I think in this case it is better to err on the side of information.

26. **P19. Section 4. The second presentation suggestion I have, is to display OCS as a VMR on a fixed altitude (altitude of max sensitivity?). Currently it is very difficult to interpret the columns over land because of orography. The maps now basically look like earth surface ground height maps. Satellites are of course sensitive to a**

**column rather than vmr at a given location, still as only one parameter is retrieved and the whole profile is scaled uniformly, it really doesn't harm to show the value at one given altitude (even though care much be taken not to over-interpret those values of course). Orographic effects should be far less visible that way and in addition it would also be much clearer whether the retrieval sees an enhancement or depletion with respect to the a priori (the apriori could be indicated on the colorbar).**

I agree with the stated drawbacks of switching the visualization to an effective VMR at maximum sensitivity: Which are that an effective VMR is NOT what is being retrieved and that over-interpretation is possible because it misrepresents the estimated quantity (i.e., total column). I disagree that switching to one altitude is harmless and that over-interpretation is not only possible, but inevitable. For example, the stated specification that IASI was built for requires temperature sounding to within 1K at a tropospheric vertical resolution of 1km. This is only possible with heavy reliance upon a priori constraints simply based on the vertical width of the Jacobians themselves. As a result, it is quite easy to assume that the vertical temperature profile estimates are independent measurements, which they are not at the presented resolution. Subsequently, such estimates require that the averaging kernels be included with the data to provide meaningful results for implementation in chemical transport models. Thomas von Clarmann wrote a paper on removing a priori from retrieved quantities and made the argument that the community should avoid this practice simply for smoother plots and figures and think ultimately about how the results will be consumed. Applying this reasoning to the OCS estimates presented, if we insist that the results must look smoother for presentation purposes while implying that vertical resolution is greater than what is physically achievable by IASI, then we build unrealistic expectations on the data consumer side that trickles through to conclusions about physical mechanisms. Even if this is explained in great detail, like averaging kernels are, the majority of readers not familiar with retrieval theory will still make the conclusion that vertical resolution is greater than it truly is. Therefore, we must accept inconvenient features, such as orography, and present the closest representation of what is being estimated independent of prior assumptions. Furthermore, the vertical profile of OCS is simply not known well enough to follow the referee's comment accurately, which leads to the next comment below.

27. **P19. Section 4. Thirdly, it would be nice to show a modeled plot of OCS vmrs to represent 'the state of the art' of the current knowledge on OCS distributions. This**

**would greatly ease discussion (it could first be discussed in section 2, and then referred to in section 4).**

Agreed, however I don't believe chemical transport models represent OCS well enough to address this comment. The RTTOV profiles used here assume a constant tropospheric VMR, which is stated in the paper. This is also what Le Kuai assumed for the TES retrieval. The leading model regarding OCS is likely the one described by Launois from Bordeaux. I attempted to contact the author via email for such profiles, but did not establish contact. Perhaps this paper will inspire further collaboration. We are already working with the University of Leicester for improved OCS modeling, but this will have to be a separate paper.

28. **P20. L4. "likely due". This would be very easy to check no? In fact, it wouldn't be too hard, and quite instructive to produce a map which for each place on Earth shows the filter which was most often applied.**

   Instead of "likely," this phrase was changed to be more definitive. The AVHRR cloud flag does routinely identify sea ice as cloud. The map suggested in this comment is essentially the number of pixels per bin map that has been included as the middle row in Figures 10-16. From this you can see which areas of the globe are frequently flagged as problematic. A further breakdown specific to cloudiness is not within the scope of this paper.

29. **P20. Section 4.1. One thing that occurred to me was that the daytime ocean seems to have higher highs and lower lows, can you confirm/explain?**

   At this early stage in the research, all that can be said about features like this are qualitative speculations. I see a few areas where the daytime oceans have higher highs, but I don't share the same opinion that the lows are noticeably lower. Whether this effect is a physical property of OCS or due to a systematic error, like solar influence, it cannot be determined without more in situ data or improved models of OCS to compare.

30. **P21 L4. Please add a number/point, as was done for the other points of interest.**

   A point number was added and the remaining increments were updated accordingly.

31. **P21/22. I would personally be even more cautious in over-interpreting the data, given the maturity of the product.**

   Point taken. Effort was put into making sure that definitive conclusions were not made and the possibility of retrieval errors are discussed. This subsection begins with a paragraph disclaimer about not overdrawing conclusions.

32. **P23. L14. "import" should probably be "important"**

   Yes, this was corrected in the paper.

**Comments from the second referee:**

33. **pg 2, L 5 Consider clarifying to indicate that the random error associated with the mean or median value can be improved by averaging.**

    A phrase about reducing the random error of the retrieval was inserted as recommended.

34. **pg 3, L 27 where you mention "manned space flight" would be good to mention the mission name.**

    The ATMOS experiment is now mentioned as recommended.

35. **pg 5, L 14 would say here that $\varepsilon$ is the error in measured signal relative to the linearised forward model rather than just forward model.**

    The word 'linearised' was added as recommended.

36. **fig 2 pg 8 are these two plots the different surface temperature contrast scenarios? Needs mentioning in the figure caption.**

    Yes, the thermal contrast conditions are now specified in the caption.

37. **pg 12, L 5 Perhaps you mean "will show an associated spectral feature beyond the standard deviation" rather than "in the standard deviation" here?**

    It is meant as stated. Figure 6 is a plot of the sample standard deviation of the residual spectrum.

38. **pg 13, L7 Perhaps the part about how you deal with spectral variations due to H2O in the measurement error covariance could be illustrated by an equation? For the nomenclature, here do you really mean the "variances of the measurement covariance" or do you mean both the covariance and variance entries of that matrix. Where it states "measurement covariance", this strictly refers to < yyT > i.e., the covariance of the measurement rather than the "measurement error covariance" < εεT > although you don't always see them distinguished in the literature. Here do you strictly mean measurement error covariance?**

    I think in this case formulating an equation will actually make the technique less understandable. Simply, the diagonals of the measurement error covariance are scaled by the ratio of the two lines plotted. Yes, measurement error covariance is meant here, specifically the diagonals of these covariances. Two sentences were slightly reworded to make this point clearer to the reader.

39. **pg 16, L5 Here it's that are only 80 independent atmospheres that implies there should be colinearilty in the overdetermined system in Eq 12 which means you expect to be able to reduce the dimensionality. Therefore perhaps reconsider the wording of the sentence "However, since there are only 80 independent atmospheres considered, the dimensionality of the problem must be reduced....".**

A comment was inserted about the problem actually being underdetermined, which is due to the colinearity as mentioned.

40. **pg 17 Found it somewhat difficult to follow which methods were being referred to in the discussion about the method to select the initial atmosphere. Perhaps you could specifically label the methods to help the reader follow more easily.**

Added annotations to the description referencing the method number as enumerated in the paper.

**Responses to Le Kuai were posted in the open discussion forum.**

**Responses to Italia De Feis:**

To the first comment about acknowledging work done by Luizzi et al in the 2016 paper. Thank you for pointing the paper out. It will be referenced accordingly as an example of an iterative retrieval of OCS. However, I think this comment really misses the thrust of this paper and why it is important. All comments and descriptions made are in the context of "fast linear retrievals." The procedure by Luizzi (while significant) is completely the opposite of the approach presented here. Luizzi et al use every single spectral channel of IASI to iteratively retrieve a large state vector of atmospheric quantities. It would be hard to devise an approach that is more computationally intensive than this. Hopefully, a global seasonal product of OCS will be computed. However, I suspect this will require a very large cluster of computer nodes and significant spatial and temporal averaging to generate this result. On the contrary, the method presented here is certainly less accurate, but orders of magnitude faster. The second paragraph specifically addresses this point. Let's not try to compare apples to oranges.

With regards to the second comment in the following paragraph, I am not sure why the phrase "unprecedented features" is in quotes. This phrase is not used anywhere in the paper. We certainly do not make the claim that this paper is the first to utilize an ensemble of prior states for atmospheric sounding. The RTTOV ensemble (which is cited) is indeed a subset of the TIGR ensemble mentioned. The following comment about jointly retrieving quantities falls along this same argument. Nowhere is this paper do we claim to have invented joint retrievals. I believe this is obvious. Specifically, this paper discusses fast linear retrievals. Within this topic, the current method most commonly published retrieves a single estimate of state where systematic errors are folded into the measurement error covariance and treated as effective noise (as discussed). Therefore, the method presented here is uncommon in that it is a fast linear retrieval that handles interfering signals via the joint estimate and selects an initialization point from an

ensemble based solely on the observed signal and pre-calculated ensemble radiances. Using ECMWF data to initialize an iterative retrieval, as mentioned in the comment, is a smart choice. However, in a fast linear retrieval this quickly becomes a speed-limiting factor. While this paper certainly does not claim to have invented the sub-components themselves, the compilation of these techniques towards this specific problem was stated as innovative by the first referee.

Next, the claim was made that training an EOF regression is similar and, furthermore, superior to the method presented in this paper. Firstly, an EOF regression technique is quite different from the method presented here. I will not go into a detailed description of EOF regressions, but in short I recall that they are statistical in nature based on using the training ensemble to save time computing the EOFs for each observation. If we pursue comparing this method to an EOF regression, then we should also compare it to an Artificial Neural Network (ANN) technique. This is outside of the scope of this paper and is best left for subsequent research. Secondly, I disagree that an EOF regression is obviously superior to the technique presented here. Like all methods, an EOF regression has its strengths and weaknesses. EOFs are ideal for reducing null dimensions. While this helps improve the condition of the inverse problem, it is not the only point to consider when performing retrievals. However, I absolutely encourage another group to try a fast EOF regression technique for OCS and generate seasonal maps for comparison.

Further responses to short comments:
1. Yes, OCS is strongly correlated to surface temperature (see Figure 2), specifically thermal contrast. However, all tropospheric gases are strongly correlated to thermal contrast and this point should be well known within the community. Providing the exact correlation value for an atmospheric scenario does not add much to this point in my opinion.
2. The wavenumber at which solar influences become noticeable above the noise level depends upon many factors, such as atmospheric transmission, angle of solar incidence, surface reflectance, and the surface BRDF. Avoiding IASI views within 18 degrees from the solar specular reflection path was not chosen carelessly, but this is the angle at which there were no noticeable changes in the OCS estimates. Furthermore, pixels that may be contaminated by solar radiances are likely screened out by the quality check on the projected cost. Certainly modeling solar radiance would be an improvement as commented and can be attempted in future research.
3. Yes, surface emissivity is a concern as stated in the paper. Please read the following sentences of the paragraph mentioned that discuss this further. Additionally, the projected cost technique does a good job of filtering out areas that may be poorly modeled. Notice

that the areas of concern (e.g., desert regions) tend to have far fewer retained pixels than areas of higher surface emissivity, like over water and dense vegetation.

4. How accurate is the Reference Forward Model? Please see the referenced paper on the RFM and its subsequent references for specifics on its accuracy. Currently $CO_2$ and $CH_4$ line mixing within Q-branches is a documented area on schedule for upgrade based off the latest spectral databases.

5. The entirety of Section 3.4 is included to show that the state vector was chosen based on representing model radiances to IASI noise levels with converging iterations rather than making an ad hoc guess. The numerous variants of other lesser quality state vectors and constraints are not shown because they do not add to the discussion.

6. Finally, the other estimates of the state vector are not shown because, as mentioned, they are only included in so far as to provide better OCS estimates. The selected channels are optimized for an OCS retrieval and not for a total joint retrieval. Therefore, the CO and O3 estimates are not deliverable products. However, they can possibly be used for quality filtering. If they were supposed to be deliverable products, then the spectral channels would have been chosen based on the combined variances of the posterior covariance matrix.

[revised manuscript text omitted]